# Imaging NRF2 activation in non-small cell lung cancer with positron emission tomography

Hannah E. Greenwood[1], Abigail R. Barber [1], Richard S. Edwards[1], Will E. Tyrrell[1], Madeleine E. George[1], Sofia N. dos Santos[1], Friedrich Baark[1], Muhammet Tanc[1], Eman Khalil[1], Aimee Falzone[2], Nathan P. Ward [2], Janine M. DeBlasi[2], Laura Torrente[2], Pritin N. Soni[2], David R. Pearce [3,4], George Firth[1], Lydia M. Smith[1], Oskar Vilhelmsson Timmermand[1], Ariana Huebner [3,4], Charles Swanton [3,4], Robert E. Hynds [3,4], Gina M. DeNicola [2] & Timothy H. Witney [1] ✉

Mutations in the NRF2-KEAP1 pathway are common in non-small cell lung cancer (NSCLC) and confer broad-spectrum therapeutic resistance, leading to poor outcomes. Currently, there is no means to non-invasively identify NRF2 activation in living subjects. Here, we show that positron emission tomography imaging with the system $x_c^-$ radiotracer, [18F]FSPG, provides a sensitive and specific marker of NRF2 activation in orthotopic, patient-derived, and genetically engineered mouse models of NSCLC. We found a NRF2-related gene expression signature in a large cohort of NSCLC patients, suggesting an opportunity to preselect patients prior to [18F]FSPG imaging. Furthermore, we reveal that system $x_c^-$ is a metabolic vulnerability that can be therapeutically targeted with an antibody-drug conjugate for sustained tumour growth suppression. Overall, our results establish [18F]FSPG as a predictive marker of therapy resistance in NSCLC and provide the basis for the clinical evaluation of both imaging and therapeutic agents that target this important antioxidant pathway.

To provide defence against oxidative stress and maintain redox homoeostasis, cells upregulate a myriad of different detoxifying enzymes and transporters. This antioxidant programme is mediated in part by transcription factor nuclear factor-erythroid 2 p45-related factor two (NRF2, *NFE2L2*). Under normal conditions, NRF2 protein levels are controlled by its negative regulator, Kelch-like ECH-associated protein 1 (KEAP1). KEAP1 binds to NRF2 in the cytosol, recruiting the Cullin-3 (Cul3) E3 ubiquitin ligase complex, which ubiquitinates NRF2, leading to its proteasomal degradation. Under conditions of oxidative stress, elevated levels of reactive oxygen species (ROS)

oxidise key cysteine residues in KEAP1 (C226, C273 and C288), altering its conformation, thereby impairing the degradation of NRF2. Newly translated NRF2 subsequently accumulates in the cell[1].

Cancer cells exploit NRF2 activation as a pro-survival technique to overcome insult from ROS[2,3]. NRF2 stabilisation is facilitated by either gain of function mutations in NRF2 or loss of function mutations in *KEAP1*. Additionally, epigenetic alterations, including CpG *KEAP1* promoter hypermethylation, mutations in the Cul3 ligase complex and oncogene-dependent transcriptional upregulation, alter KEAP1 function[3,4]. Following its stabilisation, NRF2 directly upregulates

[1]School of Biomedical Engineering & Imaging Sciences, King's College London, St Thomas' Hospital, London, UK. [2]Department of Metabolism and Physiology, H. Lee Moffitt Cancer Center, Tampa, FL, USA. [3]CRUK Lung Cancer Centre of Excellence, UCL Cancer Institute, University College London, London, UK. [4]Cancer Evolution and Genome Instability Laboratory, The Francis Crick Institute, London, UK. ✉e-mail: tim.witney@kcl.ac.uk

enzymes that control glutathione (GSH) biosynthesis; enzymes involved in phase 2 drug metabolism, such as NAD(P)H quinone oxidoreductase 1 (NQO1) and glutathione S-transferase (GST)[5]; drug efflux pumps, including ATP-binding cassette (ABC) transporters and multidrug resistance (MDR) proteins[6]; homologous DNA repair proteins[7]; and anti-apoptotic proteins, such as Bcl-2[8]. Unsurprisingly, hyperactivation of NRF2 results in resistance to a broad range of traditional and emerging therapies, including chemotherapy[9], radiotherapy[10], immunotherapy[11] and, as shown more recently, escape from KRAS G12C inhibitor treatment[12].

Activating mutations in NRF2/KEAP1 are found in approximately a third of non-small cell lung cancer (NSCLC) patients[13,14], with NRF2 activation associated with a worse prognosis in patients treated with first-line therapy[3,15]. Ten-year survival rates for NSCLC are stuck at ~5%[16], meaning there is an urgent clinical need to identify patients with poor prognosis and develop therapeutics that exploit specific vulnerabilities in these aggressive tumour subtypes. One of the >200 cytoprotective proteins directly regulated by NRF2 is system $x_c^-$[17]. System $x_c^-$ is a heterodimeric transporter that consists of two subunits: the functional transporter and light chain xCT (*SLC7A11*), which confers substrate specificity, and CD98hc (*SLC3A2*), the membrane protein common to many amino acid transporters which is responsible for membrane localisation and activity[18,19]. System $x_c^-$ functions to exchange the intracellular amino acid glutamate for extracellular cystine, which, following reduction to cysteine, provides the rate-limiting precursor for de novo biosynthesis of GSH[20], the body's most abundant antioxidant.

We have previously evaluated the positron emission tomography (PET) radiotracer (*S*)−4-(3-18F-fluoropropyl)-L-glutamate ([18F]FSPG) as a specific measure of system $x_c^-$ activity in living subjects[21–25]. In this work, we show that tumour-intrinsic KEAP1/NRF2 mutations and subsequent pathway activation can be non-invasively imaged by [18F]FSPG PET imaging in a variety of animal models of NSCLC. In mice bearing NRF2-high tumours, treatment with an antibody-drug conjugate (ADC) targeting system $x_c^-$ suppressed tumour growth and increased survival compared to mice treated with cisplatin. Taken together, our study reveals that NRF2 activation provides a metabolic susceptibility for targeted imaging and treatment.

## Results

### NRF2 activation in NSCLC cells increases system $x_c^-$ activity and [18F]FSPG cell retention

*SLC7A11* is a NRF2-regulated gene which encodes the glutamate/cystine antiporter xCT, a key component of system $x_c^-$. Here, we examined the role of NRF2 activation in relation to system $x_c^-$ activity, downstream antioxidant capacity and retention of the system $x_c^-$ substrate [18F]FSPG in NSCLC cells grown in culture (Fig. 1a). Cell lines harbouring functional *KEAP1* mutations had elevated NRF2 protein expression and increased expression of downstream NRF2 targets compared to cells with wild-type (WT) or silent *KEAP1* mutations (H23; Fig. 1b). In these NRF2-high cells, xCT was elevated compared to NRF2-low cells, which was accompanied by a 7.6-fold increase in cystine consumption (8008 ± 2827 nmol/h/100,000 cells vs. 1051 ± 3208 nmol/h/100,000 cells, respectively; $n = 4$ cell lines per group; $p = 0.01$; Fig. 1c). Conversely, intracellular glutamate concentrations were halved in NRF2-high cells compared to NRF2-low cells (2.0 ± 0.42 nmol/mg protein vs 3.9 ± 1.1 nmol/mg protein, respectively; $n = 4$ cell lines per group; $p = 0.02$; Fig. 1d), providing evidence that not only xCT expression but transporter activity were increased in NRF2-high NSCLC cells. Both glutamate and cystine are substrates for GSH biosynthesis. In cells with functional *KEAP1* mutations, GSH was increased three-fold compared to NRF2-low cells (1.67 ± 0.46 nmol/mg protein vs. 0.57 ± 0.16 nmol/mg protein, respectively; $n = 4$ cell lines per group; $p = 0.004$; Fig. 1e). Furthermore, ROS levels were more than halved in NRF2-high cells compared to NRF2-low cells ($n = 4$ cell lines per group;

$p = 0.0003$), indicating increased antioxidant capacity in cells that lack functional KEAP1 (Fig. 1f,g).

To investigate whether [18F]FSPG provides a functional readout of NRF2 status, [18F]FSPG retention was evaluated in the panel of eight NSCLC lines. Following 60 min incubation, [18F]FSPG retention was 2.6-fold higher in NRF2-high lines compared to NRF2-low lines (11.7 ± 2.8% radioactivity/mg protein vs. 4.4 ± 1.1% radioactivity/mg protein, respectively; $n = 4$ cell lines per group; $p = 0.003$; Fig. 1h). [18F]FSPG cell retention is a product of both radiotracer uptake and efflux, as [18F]FSPG can pass bidirectionally across xCT. Although net retention of [18F]FSPG was higher in NRF2-high cells, [18F]FSPG efflux was also increased in these cells (Supplementary Fig. 1), indicating increased system $x_c^-$ activity compared to NRF2-low cells. Importantly, there was a strong correlation between [18F]FSPG retention and intracellular GSH across all cell lines (R$^2$ = 0.89; $p = 0.0004$; Fig. 1i), linking [18F]FSPG retention to the NRF2-mediated antioxidant programme.

### [18F]FSPG retention is altered following pharmacological and genetic manipulation of NRF2

To better understand the relationship between NRF2 activity and [18F]FSPG retention, we treated NRF2-low cell lines with the potent ($K_d \sim 1.3$ nM) and highly specific KEAP1 inhibitor, KI696 (Fig. 2a). Through its binding to the Kelch domain of KEAP1, KI696 disrupts KEAP1-mediated proteasomal degradation of NRF2, increasing NRF2 expression and transcription of its target genes[26,27]. As expected, NRF2 activation by KI696 in NRF2-low cell lines increased the expression (Fig. 2b) and activity of system $x_c^-$ (Fig. 2c), whilst decreasing intracellular glutamate in H1975 and H23 cells (Fig. 2d). Importantly, KI696 treatment significantly raised intracellular GSH concentrations (Fig. 2e) and increased [18F]FSPG retention in comparison to vehicle controls in all but H1650 cells, where [18F]FSPG retention in treated cells was more variable (Fig. 2f).

Next, we used A549 (KEAP1 mutant) and H1299 (KEAP1 WT) cells genetically manipulated to alter NRF2 expression levels (Fig. 2g). In agreement with our previously published work[28], knockout (KO) of NRF2 in A549 cells reduced xCT expression, which was restored through ectopic expression of NRF2 (KO-R). Introducing the NRF2$^{T80K}$ mutation to H1299 cells increased NRF2 expression in H1299 cells compared to empty-vector controls (PLX317), which resulted in a less-pronounced increase in xCT. In A549 NRF2 KO cells, GSH was reduced by 73% ($p = 0.013$, $n = 4$ independent experiments), which corresponded to an 82% decrease in [18F]FSPG retention compared to WT A549 cells (1.3 ± 0.3% radioactivity/mg protein vs. 7.2 ± 0.9% radioactivity/mg protein, respectively; $n = 3–4$ independent experiments; $p = 0.0001$). Reinsertion of functional NRF2 (A549 NRF2 KO-R) restored GSH levels to baseline ($n = 4$ independent experiments; $p = 0.75$ for A549 vs. A549 KO-R; Fig. 2h) and rescued [18F]FSPG retention (6.2 ± 1.2% radioactivity/mg protein; $n = 3$ independent experiments; $p = 0.34$; Fig. 2i). Conversely, the gain-of-function NRF2 mutation (T80K) in H1299 cells failed to increase intracellular GSH (1.9 ± 0.6 nmol/mg protein in empty vector vs. 2.2 ± 1.1 nmol/mg protein in cells with NRF2$^{T80K}$; $n = 3–4$ independent experiments; $p = 0.33$; Fig. 2j). A small but significant increase in [18F]FSPG retention, however, was measured in NRF2$^{T80K}$ compared to NRF2$^{WT}$ H1299 cells (5.2 ± 0.8% radioactivity/mg protein vs. 3.7 ± 0.2% radioactivity/mg protein, respectively; $n = 3$ independent experiments; $p = 0.028$; Fig. 2k), consistent with our prior finding that NRF2$^{T80K}$ elevates cystine consumption in these cells[28].

### [18F]FSPG PET is a sensitive, non-invasive marker of NRF2 status in vivo

To investigate whether [18F]FSPG could be used as a non-invasive marker of NRF2 expression in vivo, H460 FLuc (NRF2-high) and H1299 FLuc (NRF2-low) NSCLC cell lines were orthotopically grown in the lungs of mice. Initial tumour engraftment and qualitative

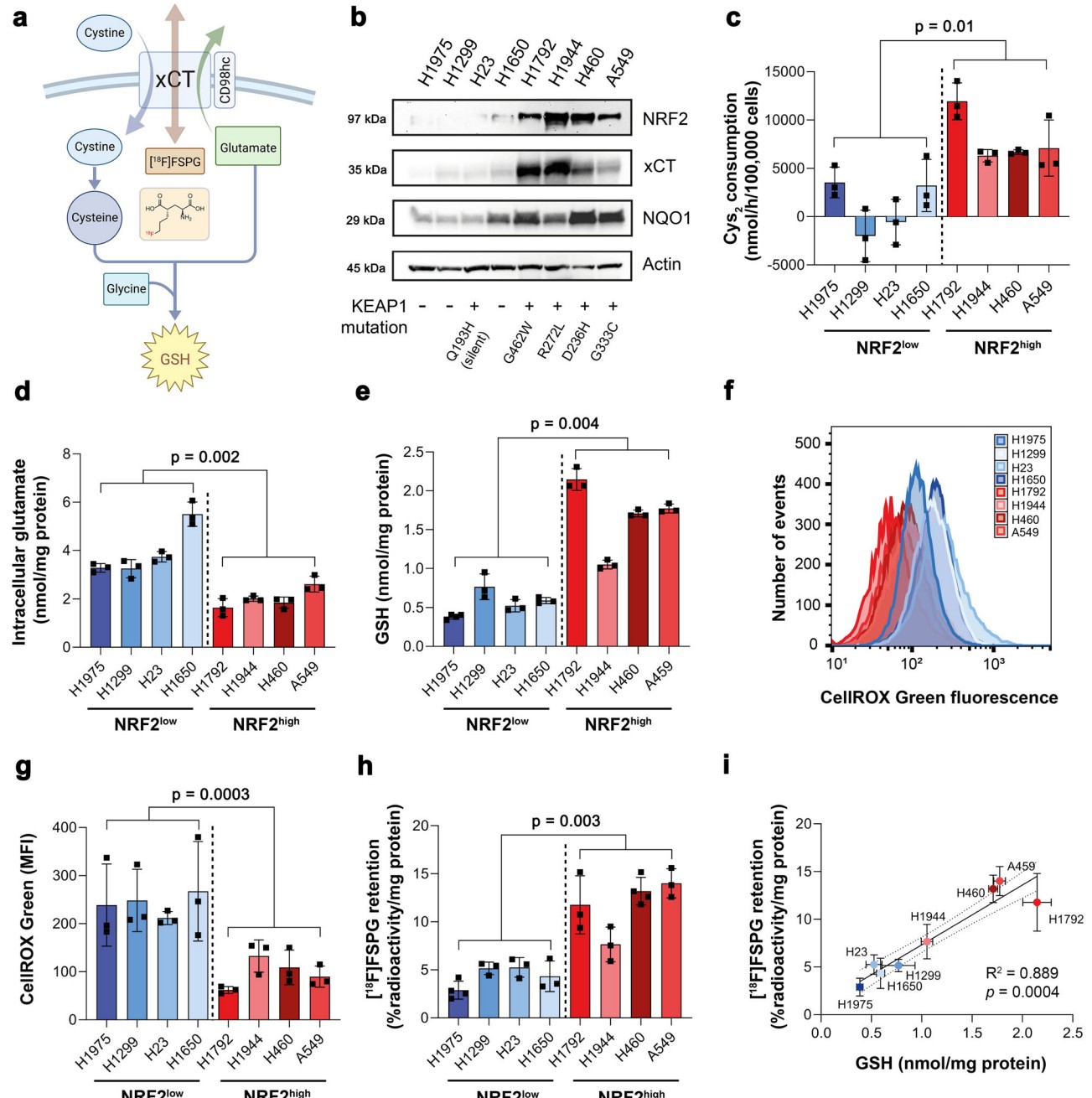

**Fig. 1 | Elevated NRF2 increases xCT expression, system $x_c^-$ activity, and downstream antioxidant capacity, detectable by [18F]FSPG. a** Schematic of system $x_c^-$ with its natural substrates cystine and glutamate, and the radiotracer [18F]FSPG (structure shown in insert). **b** Protein expression of NRF2, xCT and NQO1 in a panel of NSCLC lines and corresponding KEAP1 mutations. Actin was used as a loading control. **c** Cystine consumption in NSCLC lines following media replenishment. $Cys_2$, cystine. Intracellular glutamate (**d**) and GSH (**e**) in NSCLC lines. Flow cytometric measurement of total ROS levels using CellROX Green with representative histograms (**f**) and median fluorescent intensity (MFI; **g**) shown.

**h** Intracellular retention of [18F]FSPG. **i** Correlation between intracellular GSH and intracellular [18F]FSPG accumulation. Broken lines represent the 95% confidence interval of the best-fit line. Data are presented as mean ± SD from $n = 3$ independent experiments. Comparisons were made across the mean of $n = 4$ cells per group (NRF2-high vs. NRF2-low) using an unpaired two-tailed Student's t-test for (**c**–**e**, **g**–**h**). **a** created with BioRender.com released under a Creative Commons Attribution-NonCommercial-NoDerivs 4.0 International license, https://creativecommons.org/licenses/by-nc-nd/4.0/deed.en. For (**b**–**e**, **g**–**i**), source data are provided as a Source Data file.

assessment of tumour growth was monitored by bioluminescence imaging (BLI; Supplementary Fig. 2) and imaged by [18F]FSPG PET when lesions became visible on CT (Fig. 3a). Histological analysis of excised lungs revealed multi-focal disease and variable tumour sizes for both H1299 and H460, with substantially higher xCT expression in H460 compared to H1299 tumours (Fig. 3a). PET imaging revealed a typical pattern of [18F]FSPG distribution, characterised by low physiological uptake in all healthy organs except the pancreas

and elimination via the urinary tract (Fig. 3b and Supplementary Fig. 3). [18F]FSPG retention in both tumours was clearly visible above background (Fig. 3b), with image-quantified [18F]FSPG retention ~3-fold higher in H460 tumours compared to H1299 tumours (8.3 ± 1.6% injected dose (ID)/g protein vs. 2.8 ± 1.5% ID/g protein; $n = 7$-21 lesions from 3–9 mice; $p < 0.0001$; Fig. 3c). NRF2 was substantially higher in H460 lesions compared to H1299; a pattern that was replicated with xCT (Fig. 3d).

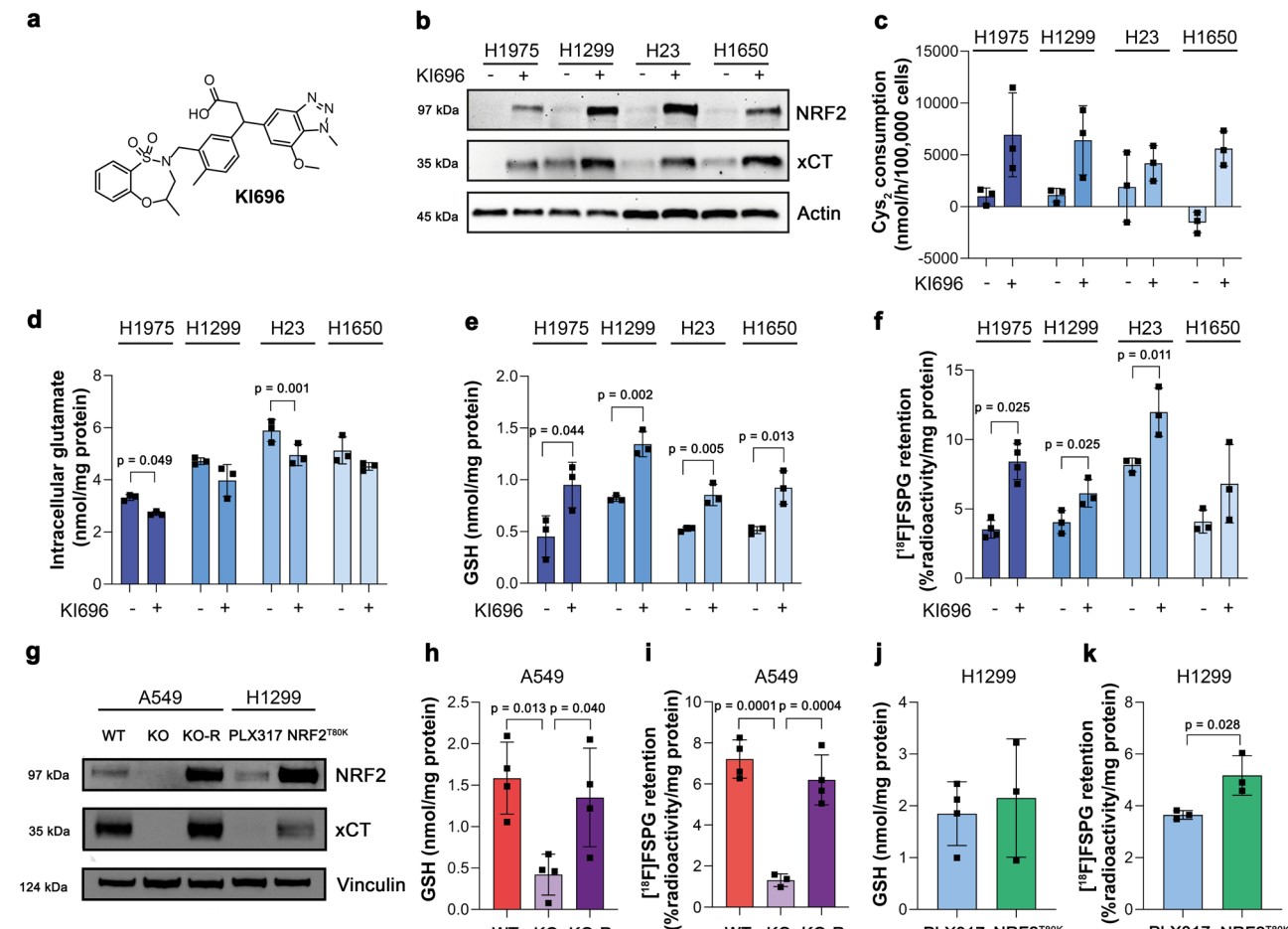

**Fig. 2 | [¹⁸F]FSPG retention is altered following pharmacological and genetic manipulation of NRF2. a** Chemical structure of KI696. **b** Representative western blot of NRF2 and xCT expression in NRF2-low cell lines 24 h post treatment with vehicle control or 200 μM KI696. Actin was used as a loading control. **c–f** Analysis of cystine (Cys₂) consumption (**c**), intracellular glutamate (**d**) and intracellular GSH (**e**) in NRF2-low lines following KI696 treatment compared to vehicle control. **f** Intracellular [¹⁸F]FSPG retention in NRF2-low cells after KI696 treatment compared to vehicle control. **g** Representative western blot of NRF2 and xCT expression

in NSCLC cells following genetic manipulation of NRF2. Intracellular GSH (**h**, **j**) and [¹⁸F]FSPG retention (**i**, **k**) in genetically modified NSCLC cells. Data are presented as mean ± SD from $n = 3–4$ independent experiments. Comparisons were made using an unpaired two-tailed Student's t-test (**d–e**, **j–k**), an unpaired one-tailed Student's t-test (**f**), or a one-way ANOVA followed by correction for multiple comparisons via the Tukey method (**h–i**). For (**b–k**), source data are provided as a Source Data file.

## [¹⁸F]FSPG retention is increased in tumours of genetically engineered mice with Nrf2 activation

To examine whether [¹⁸F]FSPG PET could identify enhanced Nrf2 activity in lung tumours of immunocompetent mice, a conditional knock-in mouse model of Nrf2^D29H/+ was used (Fig. 4a). Lung tumours were induced following intra-nasal administration of adenoviral-Cre in the Kras^G12D/+; p53^flox/flox (KP) and Kras^G12D/+; p53^flox/flox; Nrf2^D29H/+ (KPN) model[29]. Viral infection of the lungs of KP and KPN mice resulted in the development of tumours after ~3 months, with multiple lesions visible by CT (Fig. 4b). Interestingly, although the number of tumours arising from KPN mice were greater than those of KP mice (105 vs. 63), the total tumour volumes between cohorts were similar, at 22.2 ± 9.4% and 18.1 ± 6.5% of total lung volume, respectively ($n = 5–6$; $p = 0.44$; Fig. 4c). We next performed PET imaging with KP and KPN mice to non-invasively profile tumour-associated Nrf2 activity with [¹⁸F]FSPG. Representative single slice coronal [¹⁸F]FSPG PET/CT images of 40–60 min summed radioactivity are shown in Fig. 4d. [¹⁸F]FSPG retention was high in KP tumours (10.1 ± 3.3% ID/g; $n = 62$ lesions), which was further increased on average 3.6-fold in KPN tumours (36.8 ± 14.3% ID/g; $n = 104$ lesions; $p < 0.0001$; Fig. 4e). There was a heterogenous distribution of [¹⁸F]FSPG retention across KPN lesions, with some tumours reaching >80% ID/g. Following imaging, tumours

harvested from KP and KPN mice confirmed elevated Nrf2 in tumours expressing heterozygous Nrf2^D29H/+ (Fig. 4f). As expected, the Nrf2^D29H/+ mutation also increased xCT compared to Nrf2^WT tumours, both in the membrane and the cytosol (Fig. 4g, h). We previously reported that Nrf2 and the Nrf2-regulated gene Nqo1 are downregulated in high-grade tumours that arise from Keap1/Nrf2 mutant models[29]. Similarly, xCT expression was reduced in the KPN model as tumours progressed from hyperplasia (atypical adenomatous and bronchiolar) and low-grade tumours to higher grades (Fig. 4i).

## Elevated [¹⁸F]FSPG retention in NRF2-mutant tumours is recapitulated in patient-derived xenografts

Lung TRACERx (TRAcking Cancer Evolution through therapy (Rx)) is a prospective cohort study that aims to define how intratumour heterogeneity affects the risk of reoccurrence and survival in NSCLC[30]. Transcriptomic data from the TRACERx 421 cohort[31] revealed highly variable *NFE2L2* and NRF2-regulated gene mRNA abundance, both within and between tumours (Supplementary Fig. 4). There was good concordance between *SLC7A11* (which encodes xCT), *NQO1*, *GCLC* and *GCLM* median expression levels across tumours (Fig. 5a), providing an 'antioxidant signature' associated with the NRF2 transcriptional programme. We identified patient-derived xenograft (PDX) models from

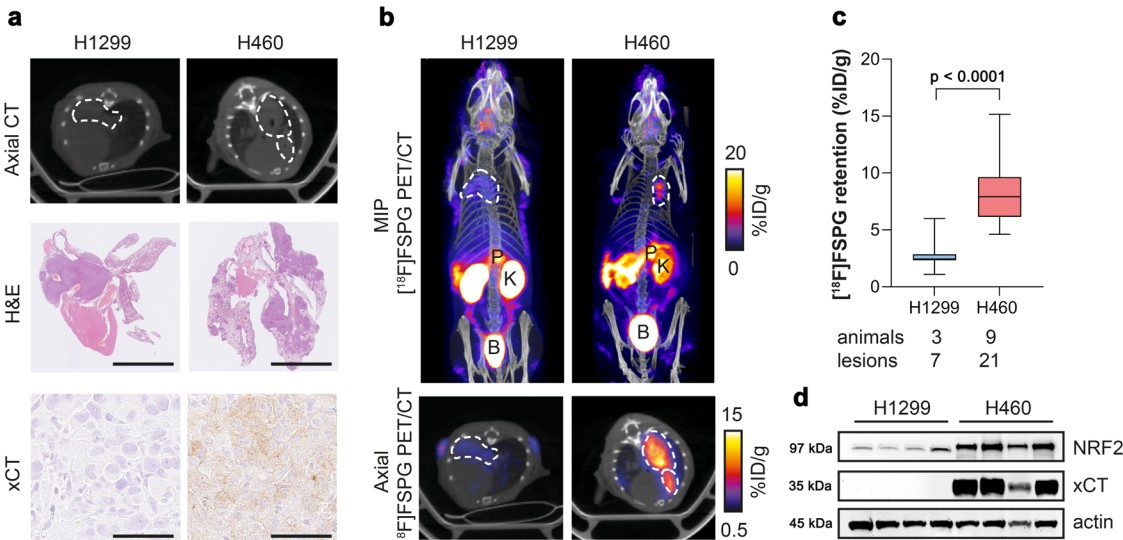

**Fig. 3 | [18F]FSPG PET can differentiate NRF2-high from NRF2-low tumours when grown orthotopically in the lungs of mice. a** Single slice CT axial images (top) and ex vivo H&E images (middle; scale bar = 5 mm) of lungs containing H1299 or H460 tumours, with xCT staining of corresponding tumours (bottom; scale bar = 50 μm). **b** Representative in vivo [18F]FSPG PET/CT maximum intensity projections (MIPs; top) and axial single-slice PET/CT (bottom) of mice bearing H1299 or H460 orthotopic lung tumours. Dashed lines represent the tumour outline. P pancreas, K kidney, B bladder. **c** Quantified [18F]FSPG retention in individual tumour lesions from orthotopic tumour-bearing mice. Comparison was made using an unpaired two-tailed Student's t-test. $n = 7$-21 lesions from 3 to 9 mice per cohort. The median value (center line), lower quartile and upper quartile (box edges) and maximum and minimum value whiskers are indicated in the boxplot. **d** Representative western blot for xCT and NRF2 expression in H1299 and H460 orthotopically grown tumours from $n = 4$ mice per group. Actin was used as a loading control. For (**c, d**), source data are provided as a Source Data file.

TRACERx patients which were either *NFE2L2* wildtype (WT; CRUK0640 region 8; R8) or contained truncal alterations in *NFE2L2* (c.G187C/ p.E63Q and c.G352C/p.E118Q; CRUK0772 R1)[32]. CRUK0640 R8 had truncal mutations in *NF1* and *KMT2D*, while CRUK0772 R1 had *TP53* and *RB1* mutations in addition to the *NFE2L2* mutations. In patient RNA sequencing data, *SLC7A11*, *NQO1*, *NFE2L2* and *GCLM* gene expression was higher in CRUK0772 tumour regions compared to CRUK0640 (Supplementary Fig. 5).

Once tumour xenografts reached 150 mm³, mice bearing NRF2 WT and mutant tumours underwent [18F]FSPG PET/CT imaging. As with orthotopic and syngeneic tumours, [18F]FSPG retention was higher in the NRF2-mutant xenografts compared to WT (Fig. 5b and Supplementary Fig. 6), at 14.8 ± 2.8% ID/g and 7.0 ± 2.4% ID/g, respectively ($n = 9$–11 mice; $p < 0.0001$; Fig. 5c). There was a high degree of intra-tumoural heterogeneity of [18F]FSPG retention for both PDX models which was not related to cellularity, as confirmed by ex vivo auto-radiography (Fig. 5d). Regional sampling of the xenograft tissue revealed high but variable xCT expression in CRUK0772 R1 xenografts that matched the pattern of NRF2 expression, whereas neither xCT nor NRF2 could be detected by western blot in CRUK0640 R8 xenografts (Fig. 5e). The elevated antioxidant capacity of NRF2-mutant xenografts was reflected in a doubling of GSH in these tumours (0.90 ± 0.46 nmol/ mg protein for CRUK0772 R1 compared to 0.43 ± 0.12 nmol/mg pro-tein for CRUK0640 R8; $n = 4$ tumours/group; $p = 0.047$; Fig. 5f), again recapitulating the pattern of [18F]FSPG retention.

## [18F]FSPG identifies NRF2 pathway activation with high sensi-tivity and specificity

To better understand the degree by which [18F]FSPG can distinguish NRF2-high from NRF2-low tumours, we analysed the areas under the receiver-operating characteristic (ROC) curve for each in vivo model. The ROC illustrates the performance of a binary classifier model at varying threshold levels, with the true positive rate (y-axis) plotted against the false positive rate (x-axis) at each threshold. The area under the curve (AUC) provides the overall performance of [18F]FSPG as a binary classifier across all thresholds. For all tumour models evaluated in this study (Figs. 3, 4 and 5), [18F]FSPG had AUC values approaching 1, at 0.973 ($p = 0.0002$), 0.996 ($p < 0.0001$) and 0.990 ($p = 0.0002$) for orthotopic, GEMM and PDX tumours, respectively (Fig. 6).

## System $x_c^-$ is a vulnerability that can be exploited for targeted therapy

ADCs are a promising class of cancer therapeutics that provide preci-sion targeting of cell surface antigens and delivery of a therapeutic payload following receptor internalisation. Given that xCT is upregu-lated in tumours with activated NRF2, xCT may provide a specific vulnerability of treatment-resistant cancer that can be exploited ther-apeutically. To test this hypothesis, we assessed the efficacy of a humanised xCT-targeting monoclonal antibody conjugated to tesirine (HM30-tesirine; Fig. 7a) in NSCLC. Conjugation of tesirine to the anti-body did not affect its stability (Supplementary Fig. 7) and HM30-tesirine selectively bound to xCT present in the lysates of H460 cells. In contrast, no binding was present in low xCT-expressing H1299 cells (Fig. 7b). HM30-tesirine induced a dose-dependent increase in cell kill in H460 cells, with an EC$_{50}$ of 3.7 nM ± 2.4 nM, whereas the NRF2 and xCT-low H1299 cells had an EC$_{50}$ of 120 nM ± 50 nM, suggesting reduced sensitivity to this ADC ($n = 3$; $p = 0.018$; Fig. 7c).

We have shown above that [18F]FSPG measures NRF2 pathway activation with high sensitivity and specificity. Next, we performed longitudinal PET imaging in mice bearing subcutaneous H460 or H1299 tumours to stratify responders from non-responders. There was a marked and statistically significant difference in [18F]FSPG retention between these two tumour types (Fig. 7d). This pattern was replicated across all tumour sizes and stages of development, establishing that even in very small tumours (20–60 mm³), [18F]FSPG can discriminate NRF2 activation status (Supplementary Fig. 8). [18F]FDG, however, was unable to differentiate between these NRF2-high and NRF2-low tumours (Supplementary Fig. 9).

Having stratified tumours as [18F]FSPG-high (H460) or [18F]FSPG-low (H1299), the in vivo efficacy of HM30-tesirine was examined in

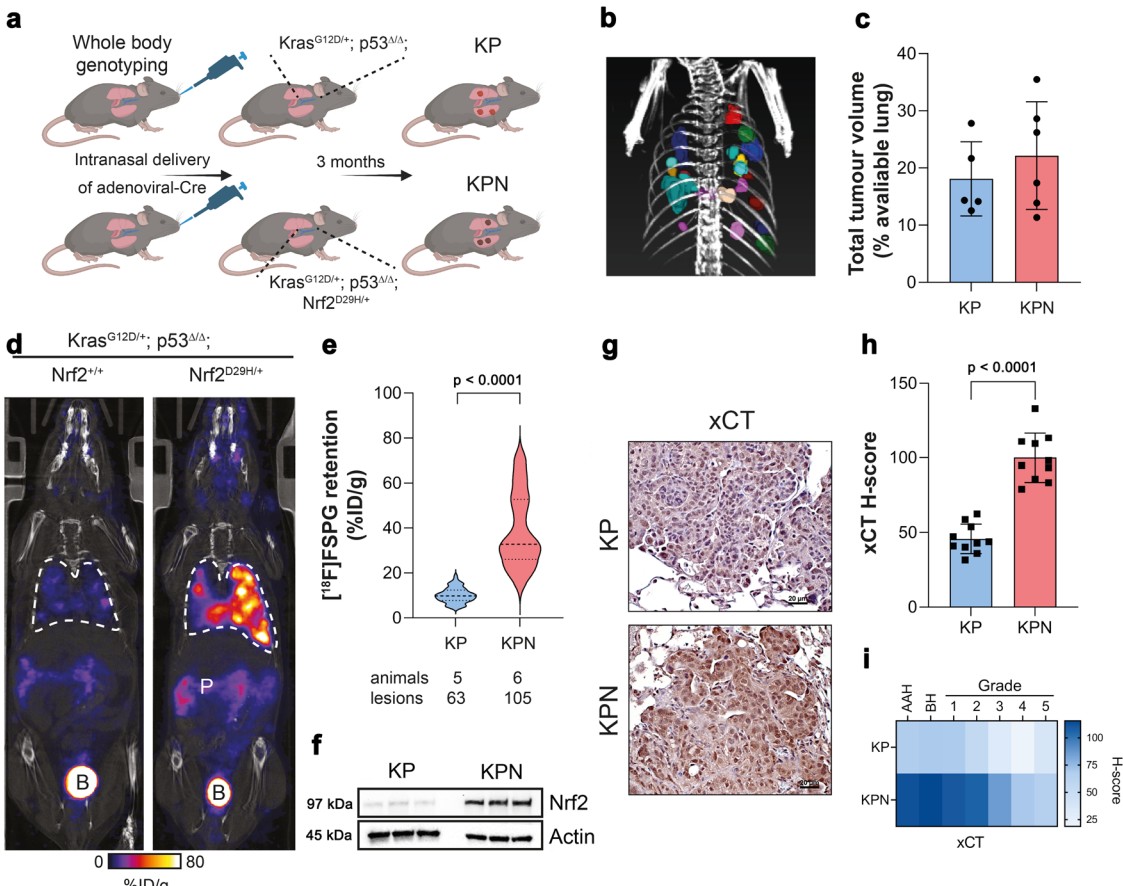

**Fig. 4 | [¹⁸F]FSPG retention is increased in Nrf2 mutant mice. a** Scheme depicting tumour formation in KP and KPN mice. KP mice conditionally express oncogenic Kras and have loss of p53 function. KPN mice conditionally express oncogenic Kras, have loss of p53 function and express a mutant Nrf2, which increases Nrf2 protein levels. **b** CT MIP representing individual 3D tumour regions of interest. **c** Total tumour volumes in the lungs of KP and KPN mice. Data are presented as mean ± SD from $n$ = 5-6 mice. **d** Representative coronal [¹⁸F]FSPG PET/CT images of 40–60 min summed activity in KP and KPN tumour-bearing mice. Dashed white lines indicate the lung. B bladder, P pancreas. **e** Violin plots of [¹⁸F]FSPG tumour retention from individual lesions. Dashed lines represent the median and the upper and lower quartiles. $n$ = 63–105 lesions from 5–6 mice per cohort. **f** Nrf2 expression in KP and KPN tumour lesions. Actin was used as a loading control. **g** Representative IHC staining of xCT from lesions taken from KP and KPN mice (scale bars = 20 μM). **h** H-scores for xCT IHC staining. Data are presented as mean ± SD from $n$ = 10 mice. **i** Heatmap depicting the H-scores for xCT IHC staining by tumour grade. AAH adenomatous atypical hyperplasia, BH bronchiolar hyperplasia. All comparisons were made using an unpaired two-tailed Student's t-test. **a** created with BioRender.com released under a Creative Commons Attribution-NonCommercial-NoDerivs 4.0 International license, https://creativecommons.org/licenses/by-nc-nd/4.0/deed.en. For (**c**, **e**, **f**, **h**), source data are provided as a Source Data file.

these tumour-bearing mice. Tumours were grown subcutaneously until they reached ~70 mm³–a point where they were well-vascularised. Mice were randomised into vehicle, cisplatin and HM30-tesirine treatment groups, with the latter receiving three intra-peritoneal doses of 1.5 mg/kg HM30-tesirine 7 days apart. No acute toxicity was associated with HM30-tesirine administration, assessed through measurements of body weight (Supplementary Figs. 10, 11) and observation of clinical symptoms.

HM30-tesirine substantially inhibited tumour growth in H460 tumours compared to vehicle and cisplatin-treated mice over the 6-week treatment study, with effects still present >20 days after the last course of treatment (Fig. 7e). The initial dose of HM30-tesirine maintained tumour volumes close to their pre-treatment size 8 days post-treatment (78 ± 31 mm³), whereas control and cisplatin-treated tumours had reached 518 ± 130 mm³ and 199 ± 72 mm³, respectively by this time ($n$ = 3–6 mice; $p$ = 0.0013 for control vs. cisplatin; $p$ < 0.0001 for control vs. HM30-tesirine; $p$ = 0.10 for cisplatin vs. HM30-tesirine). HM30-tesirine treatment extended the lives of all H460 tumour-bearing mice enroled, increasing the median survival from 8.5 days in control mice to 36.5 days following HM30-tesirine treatment ($p$ = 0.0007; Fig. 7f). Whilst cisplatin induced a small but

significant reduction in the rate of tumour growth, this standard-of-care treatment did not affect overall survival compared to control tumours (median survival = 12 days; $p$ = 0.22 for control vs. cisplatin). Conversely, HM30-tesirine did not affect tumour growth or survival in NRF2- and xCT-low H1299 tumours compared to vehicle control animals, whereas cisplatin induced a growth delay (Supplementary Fig. 11).

Immunohistochemical analysis of tumours taken at humane endpoints or the end of the 6-week study revealed a 65% reduction in H460 tumour cell proliferation in mice treated with HM30-tesirine, with 28 ± 9.4% of cells staining positive for Ki67, compared to 83 ± 8.4% in control tumours ($n$ = 3 individual tumours; $p$ = 0.0016; Fig. 7g, h). At these late time points, there was no difference in levels of apoptosis between groups, as shown by cleaved caspase 3 staining (32 ± 11% positive cells for control vs. 28 ± 5.7% for HM30-tesirine; $n$ = 3 individual tumours; $p$ = 0.66; Fig. 7g, i), presumably due to the removal of dead cells by the innate immune system.

## Discussion
Resistance to therapy is one of the biggest problems in clinical oncology. Despite a revolution in new anti-cancer therapies, such as

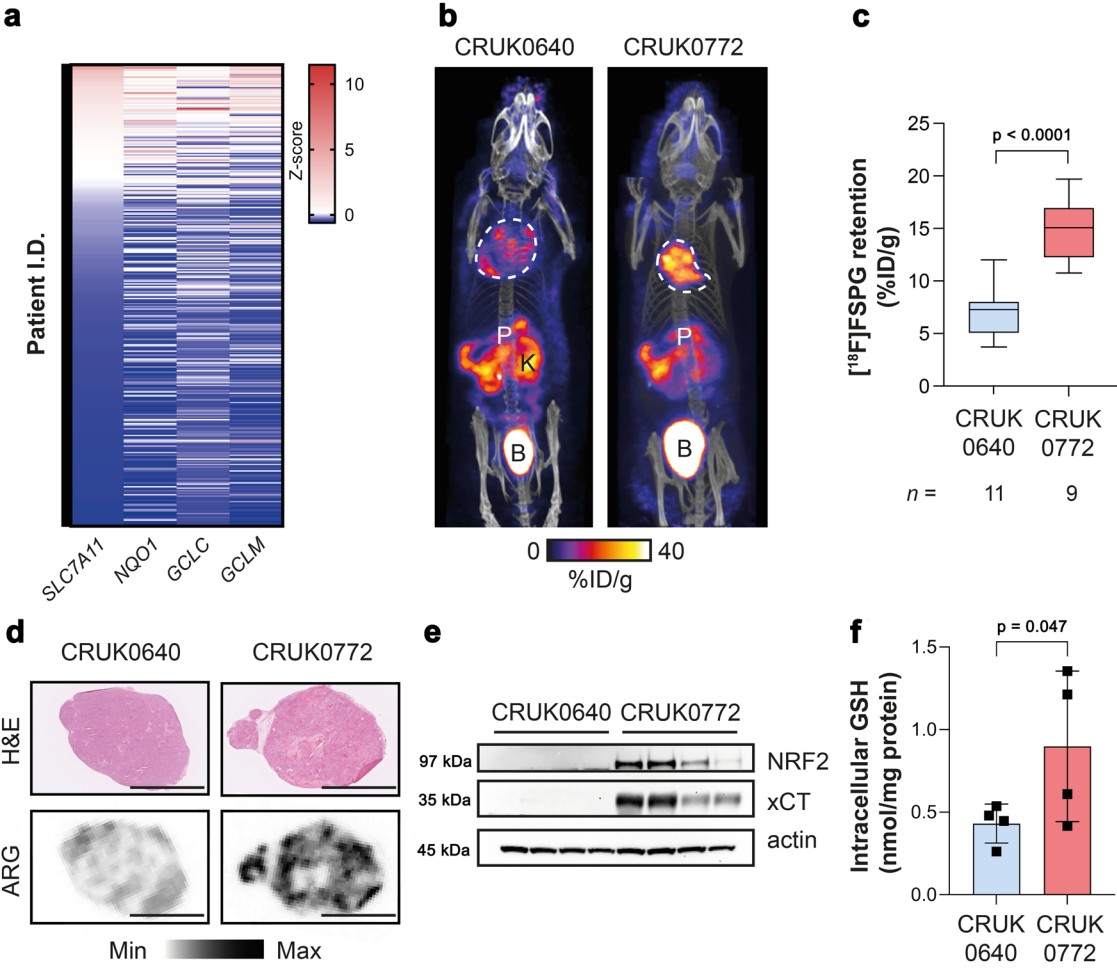

**Fig. 5 | An antioxidant gene signature accompanies NRF2 mutations in patient tumours and patient derived xenograft (PDX) models, which is detectable by [18F]FSPG PET. a** Expression of NRF2-regulated genes in the TRACERx 421 patient cohort. **b** Representative [18F]FSPG MIP of mice bearing PDXs either with (CRUK0772 R1) or without (CRUK0640 R8) a NRF2 mutation. **c** Quantification of [18F]FSPG tumour retention. The median value (center line), lower quartile and upper quartile (box edges) and maximum and minimum value whiskers are indicated in the boxplot. Comparison was made using an unpaired two-tailed Student's t-test. $n$ = 9–11 mice. **d**, Post-imaging H&E-stained tissue sections and corresponding autoradiograms (ARG) from PDXs, illustrating the intratumoural heterogeneity of [18F]FSPG retention. Scale bar = 5 mm. **e** xCT and NRF2 protein expression in PDX xenografts ($n$ = 4 regions per tumour type). Actin was used as a loading control. **f** GSH measurements from PDX tumours. Data are presented as mean ± SD from $n$ = 4 mice. Comparison was made using an unpaired two-tailed Student's t-test. For (**a, c, e, f**) source data are provided as a Source Data file.

checkpoint inhibitors and proton beam therapy, durable responses are often not observed due to acquired or innate resistance to existing treatment regimens[33,34]. Currently, there is no satisfactory way to identify patients that are refractory to therapy early on in their treatment pathway[35]. Treatment failure plays a critical role in the management of patients with NSCLC, where survival rates have struggled to improve over the last 10 years despite advances in prevention, screening and treatment[36]. In NSCLC, constitutive NRF2 activation results in resistance across the spectrum of currently available therapeutics[9–12]. A non-invasive measure of NRF2 activation may therefore provide an attractive solution for the prediction of therapy resistance in NSCLC, which may further reveal cancer-specific vulnerabilities for the precision treatment of refractory disease.

Using a combination of metabolomics, genetic engineering, drug treatment and biochemical analyses, we demonstrated the ability of system $x_c^-$ to report on NRF2 activation. In NSCLC cells grown in culture, KEAP1 loss-of-function mutations significantly increased xCT expression, had higher cystine consumption from the medium and decreased intracellular glutamate concentrations—features indicative of increased cystine/glutamate exchange by system $x_c^-$. Given that cystine influx drives GSH biosynthesis, it was unsurprising that NRF2-

high cells had elevated steady-state GSH and a concomitant reduction in ROS. Similar observations were made when KEAP1 WT cells were treated with the KEAP1/NRF2 inhibitor KI696: xCT was increased, along with GSH concentration and cystine utilisation. NRF2 depletion in KEAP1 mutant A549 cells had the opposite effect, which was rescued following ectopic expression of NRF2.

[18F]FSPG PET has been used to image system $x_c^-$ activity in a range of human malignancies, including NSCLC[37–40]. Although exclusively used in clinical trials as a diagnostic agent, we[21–25] and others[41] have shown that [18F]FSPG is a sensitive marker of therapy-induced changes in tumour redox status. Here, disruption of redox homoeostasis through NRF2 activation increased [18F]FSPG cell retention; changes that were tightly correlated to cellular GSH concentration ($R^2$ = 0.89). Interestingly, whilst the R272L NRF2 mutation in H1944 cells increased both NRF2 and xCT expression, increased GSH and [18F]FSPG retention compared to WT cells was muted, suggesting that not just system $x_c^-$ activity, but the fate of its substrates is reflected in the imaging readouts. In our genetic models and following pharmacological intervention, [18F]FSPG retention mirrored NRF2 activation levels.

Whilst other non-radioactive assays are better suited to evaluate NRF2 status in isolated cells, PET imaging facilitates the measurement

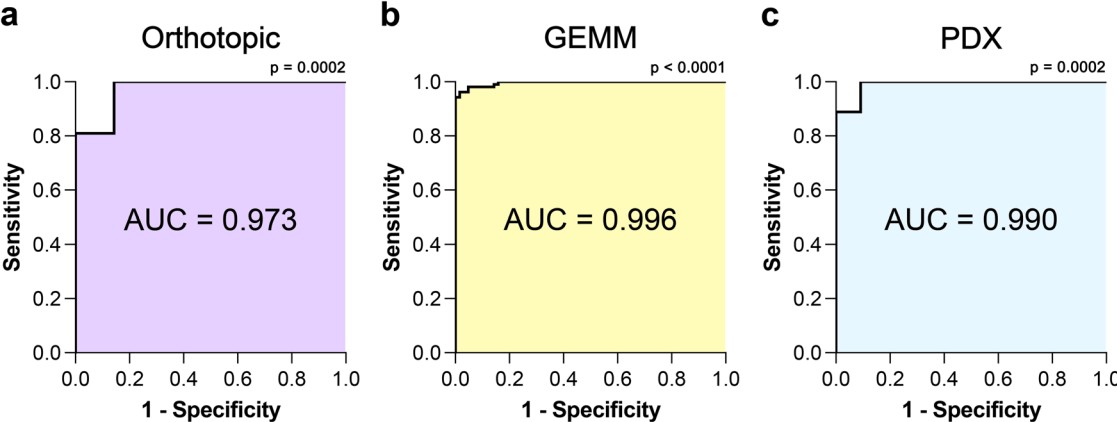

**Fig. 6 | [¹⁸F]FSPG can differentiate NRF2 activation status with high sensitivity and specificity.** ROC analysis of PET imaging data in orthotopic (**a**), GEMM (**b**) and PDX (**c**) tumours. Comparisons between the calculated AUC and AUC = 0.5 were computed from the z ratio using the equation $z = (A-0.5)/SE$, where A is the area under the curve, and SE is the standard error of the area. The P value was determined from the normal distribution (two-tail). The original source data is provided in the Source Data file for Figs. 3, 4 and 5.

of biochemical processes across the whole body, permitting assessment of the entire tumour burden[42]. This is particularly important when evaluating clonal heterogeneity, as evident in our PDX model studies (Fig. 5). The TRACERx consortium has pioneered multi-region tumour sampling and phenotyping using next-generation sequencing approaches, revealing insights into tumour evolution[30]. However, this extensive tumour characterisation, using specialised clinical and data analysis pipelines, is currently limited to a minority of patients in research studies. Alternatively, [¹⁸F]FSPG imaging could be made available to patients at most major hospitals which currently use [¹⁸F]2-fluoro-2-deoxy-D-glucose ([¹⁸F]FDG) for tumour staging/restaging. Across a range of genetically engineered, patient-derived and orthotic NSCLC models, [¹⁸F]FSPG retention increased in tumours with NRF2 activation compared to those with a normal-functioning NRF2/KEAP1 axis. The heterogeneity of this imaging signal was reflected in ex vivo markers of NRF2, xCT and GSH, providing further evidence that [¹⁸F]FSPG reports on tumour redox status. Importantly, RNA sequencing data from TRACERx patients suggested that an 'antioxidant signature' of NRF2-regulated genes could be used to select patients for further monitoring by [¹⁸F]FSPG PET imaging. Furthermore, this antioxidant signature may be applicable to other cancer types with elevated NRF2, such as pancreatic ductal carcinoma[43].

Given that NRF2 and xCT are intrinsically interconnected and NRF2 confers therapy resistance, we investigated whether xCT was a vulnerability that could be exploited to treat refractory disease. We assessed an xCT-specific ADC, HM30-tesirine, in mice bearing H460 xenograft tumours harbouring the D236H KEAP1 loss-of-function mutation. H460 tumours are therapy-resistant, partly due to NRF2 activation[44], with cisplatin treatment yielding only a moderate reduction in tumour growth rate. In these therapy-resistant tumours, HM30-tesirine treatment resulted in persistent tumour growth suppression and decreased proliferation, which was maintained following removal of treatment, likely due to the long circulation half-life of the ADC. Importantly, HM30-tesirine significantly extended the lives of H460 tumour-bearing mice compared to both cisplatin- and vehicle-treated animals. Furthermore, HM30-tesirine was ineffective in NRF2 WT tumours, highlighting the need for a non-invasive method (e.g. [¹⁸F]FSPG) to stratify responders from non-responders. Small molecule drugs that target xCT, such as imidazole ketone erastin and sorafenib, also display potent anti-tumour effects[45]. However, the relatively poor metabolic stability of these therapeutics[46], and additional mechanisms of action besides xCT inhibition[47], may preclude their future clinical

utility. This xCT ADC may circumvent some of these issues, providing a new therapeutic avenue to treat refractive NSCLC.

Despite these advances, the imaging and targeted treatment of NSCLC with high system $x_c^-$ expression holds specific challenges. [¹⁸F]FSPG is bidirectionally transported across the plasma membrane. As shown here, tumours with high NRF2 not only have elevated uptake but also increased efflux from the cell (Fig. 1h, Supplementary Fig. 1). Consequently, net [¹⁸F]FSPG in NRF2 activated tumours was increased ~3-fold compared to NRF2 WT tumours, despite orders of magnitude differences in protein expression. Moreover, the activity of system $x_c^-$ is not solely driven by transporter expression, but by the concentrations of both cystine and glutamate across the membrane[48]. Nutrient composition in the tumour microenvironment, metabolic alterations (e.g. anaplerosis), or other gene mutations (e.g. STK11) may therefore affect [¹⁸F]FSPG irrespective of NRF2 status. Antibody-based approaches (such as HM30-tesirine) are not dependent on transporter activity and may therefore simplify therapeutic strategies.

Here, we provide proof-of-concept data that imaging system $x_c^-$ may provide a means to select tumours most likely to respond to HM30-tesirine treatment. However, care should be taken in interpreting the resulting efficacy data, as the treatment was performed in a small selection of xenograft models. Future work in additional animal models is still required to demonstrate the connection between NRF2 activation and treatment efficacy before clinical translation. Additionally, system $x_c^-$, is not exclusively expressed on tumour cells, with high expression found in the brain, pancreas and components of the immune system, which may provide confounding background signal in the PET images[49,50]. Although HM30-tesirine was well tolerated in mice, patients receiving xCT-targeted therapies should be monitored for potential on-target, off-tumour toxicity. As with any other surrogate marker, system $x_c^-$ expression is also influenced by factors other than NRF2, such as ATF4, p53 and CD44v[51]. Consequently, prospective clinical trials are required to assess the specificity of [¹⁸F]FSPG for NRF2 in humans and to determine the thresholds of [¹⁸F]FSPG retention to classify tumours into NRF2-high or NRF2-low.

In summary, we describe the use of a clinically-tested PET radiotracer, [¹⁸F]FSPG, for the imaging of NRF2 activation in NSCLC. This study sets the foundation for the clinical assessment of NRF2 in NSCLC patients with [¹⁸F]FSPG at King's College London (clinical trials number: NCT05889312). If successful, the combined imaging and xCT-targeted treatment of NSCLC with activating NRF2/KEAP1 mutations may represent a new paradigm for patients with therapy-resistant disease.

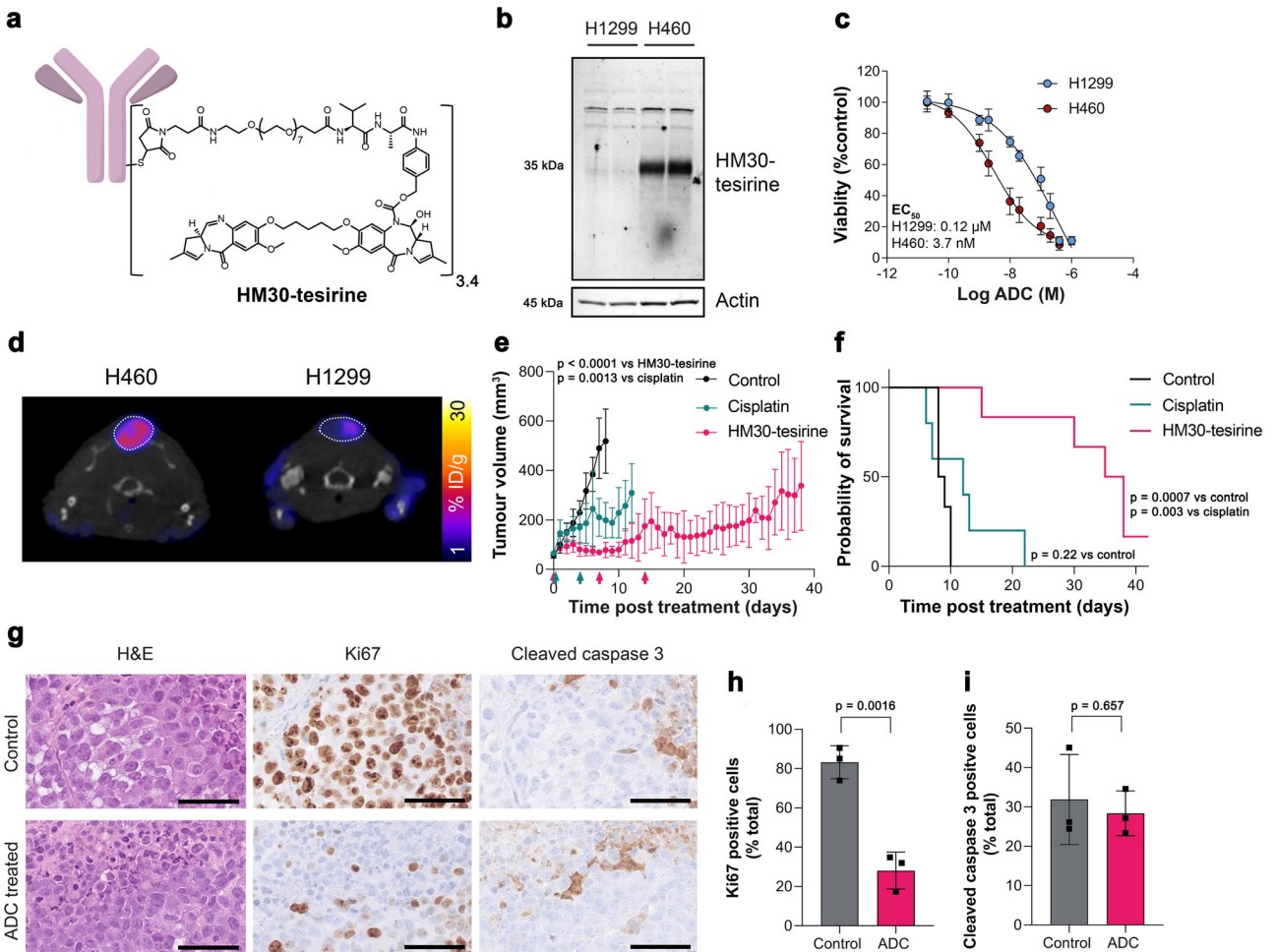

**Fig. 7 | HM30-tesirine controls tumour growth and prolongs survival of mice bearing subcutaneous H460 tumours. a** Structure of the anti-xCT tesirine conjugate, HM30-tesirine. **b** Western blot using HM30-tesirine as the primary antibody in H1299 and H460 cell lysates. Actin was used as a loading control. **c** HM30-tesirine MTT dose-response in H460 and H1299 cells. **d** Axial [¹⁸F]FSPG PET/CT images from mice bearing H460 or H1299 tumours before initiation of treatment. The dashed circle indicates the tumour. Antitumour activity (**e**) and survival benefit (**f**) of control (saline treated), cisplatin treated and HM30-tesirine treated mice bearing subcutaneous H460 tumours. *n* = 5-6 mice per cohort. The arrows under the x-axis of (**e**) represent treatment cycles for cisplatin (green) and HM30-tesirine (red), with tumour volumes measured using electronic callipers. **g** IHC for Ki67 and cleaved caspase 3 from FFPE tumours taken at endpoint. Scale bar = 50 μm. Corresponding quantification of tissue staining for Ki67 (**h**) and cleaved caspase 3 (**i**). Data in (**h**, **i**) are presented as the mean values ± SD from *n* = 3 mice. For (**c**, **h**, **i**), comparisons were made using an unpaired two-tailed Student's t-test. For (**e**) comparisons were made on day 8 using a one-way ANOVA with followed by t-tests multiple comparison correction (Tukey method). For (**f**), statistics were analysed with a log-rank (Mantel–Cox) test. To control the family-wise error rate in multiple comparisons, crude p values were adjusted by the Holm–Bonferroni method. For (**b**, **c**, **e**, **f**, **h**, **i**) source data are provided as a Source Data file.

## Methods

### Ethical approval

All animal experiments performed in the U.K. were in accordance with the United Kingdom Home Office Animal (Scientific Procedures) Act 1986 (project: PP9982297) and received local Animal Welfare and Ethical Review Body (AWERB) approval. For GEMM studies, Mice were housed and bred following the ethical regulations and approval of the University of South Florida Institutional Animal Care and Use Committee (protocol numbers: IS00005814M and IS00003893R). All animals were housed in Tecniplast Greenline Sealsafe IVC cages, fed standard rodent chow, received water ad libitum and were housed in a facility with a 12 h light cycle at 20–24 °C and 45–65% humidity.

### Cell lines and cell culture

Human NSCLC lines A549 (ATCC-CCL-185), H460 (ATCC-HTB-177), H1944 (ATCC-CRL-5907), H1792 (ATCC-CRL-5895), H23 (ATCC-CRL-5800), H1299 (ATCC-CRL-5803), H1975 (ATCC-CRL-5908) and H1650 (ATCC-CRL-5883) were purchased from LGC Ltd. H460 FLuc were purchased from Revvity (BW124316). H1299 FLuc were purchased from AMSBIO (SC053-L). A549 NRF2 knockout (KO) cells were previously generated by CRISPR/Cas9[28,52]. A549 NRF2 KO cells were reconstituted with pLX317-NRF2 (KO-restored, KO-R) by lentiviral transduction. Similarly, H1299 cells were lentivirally transduced with empty vector (pLX317) or pLX317-NRF2 T80K to generate isogenic NRF2 active H1299 cells[28]. Transduced cells were selected in 1 μg/mL puromycin for 3 days. To pharmacologically activate NRF2 expression, cells were treated with 100 nM of the KEAP1 inhibitor, KI696, for 24 h (Sigma-Aldrich, SML3618). All cells were cultured in RPMI media (Sigma-Aldrich, R0883), supplemented with 2 mM glutamine (ThermoFisher Scientific, 25030081), 10% foetal bovine serum (ThermoFisher Scientific, A5256701) and 100 U.mL⁻¹ penicillin and 100 μg.mL⁻¹ streptomycin (Sigma-Aldrich, P4333), maintained at 37 °C and 5% $CO_2$. Cell lines were tested monthly for mycoplasma infection (Eurofins) and were authenticated by Short Tandem Repeat profiling by the provider.

Cells were seeded for 24 h prior to experimental endpoint in 6-well plates in 2 mL of media. All cells were seeded at $1.5 \times 10^5$/mL,

except for H1944 and H1792, which were seeded at $2.75 \times 10^5$/mL, H1650, which was seeded at $1.75 \times 10^5$/mL and H23 and H1975, which were seeded at $2.5 \times 10^5$/mL. For KI696 treatments, H1650, H23, H1975 and H1299 were seeded at $1.0 \times 10^5$/mL, $1.0 \times 10^5$/mL, $1.25 \times 10^5$/mL and $0.75 \times 10^5$/mL, respectively in 2 mL of media 24 h prior to treatment.

## Western blotting

Western blot analysis was carried out using an iBind Flex system (ThermoFisher Scientific), according to the manufacturer's instructions[21]. For cell lysate collection, cells were seeded at densities described above for 24 h prior to harvesting. Briefly, cells were placed on ice, washed three times with ice-cold PBS and lysed in 100 μL RIPA buffer (ThermoFisher Scientific, 89901) containing 1× protease and phosphatase inhibitors (ThermoFisher Scientific, 78442). Collected lysates were then centrifuged at $15,000 \times g$ at 4 °C for 10 min. Cell debris was removed and the supernatant aliquoted to avoid freeze-thaw cycles.

For ex vivo sample preparation, tissues were dissected immediately following sacrifice, snap-frozen in liquid nitrogen and stored at −80 °C. Frozen tissues were added to pre-chilled Lysing Matrix tubes containing 1.4 mm ceramic beads (MP Biomedicals, 1169130-CF) and 1 mL RIPA buffer containing 1× protease and phosphatase inhibitors. Samples were lysed by rapid shaking using a high-speed benchtop reciprocating homogeniser, cooled to 4 °C (Precellys 24 homogenizer, Bertin Instruments). The lysates were centrifuged at $15,000 \times g$ at 4 °C for 10 min and the supernatant collected for analysis. The protein content in each lysate was determined using a BCA assay kit (ThermoFisher Scientific, 23225) and 20 μg of protein was loaded into each well.

Blots were probed for xCT (1291, lot 4), NRF2 (12721) and NQO1 (62262) at 1:1000 dilution (Cell Signalling Technology), except for Fig. 2g, where the anti-xCT antibody was from Abcam (ab37185, 1:1000 dilution). Either actin (Cell Signalling Technology, 4967) or vinculin (Sigma-Aldrich, V9264) were used as loading controls at 1:1000 dilution, with a horseradish peroxidase (HRP) linked anti-rabbit IgG secondary antibody (1:2000 dilution; Cell Signalling Technology, 7074). After antibody incubations, membranes were removed from the iBind Flex system and washed in 50 mL of tris-buffered saline with 0.1% Tween 20 (TBST) five times for 5 min on a shaker (Stuart gyratory rocker SSL3). To visualise proteins, 4 mL of Amersham™ ECL Prime Western Blotting Detection Reagent (Cytiva, RPN2232) was added to each membrane in the dark for 1 min. Images were taken using an iBright CCD camera (Invitrogen) or x-ray film. When using the iBright, images were always acquired within the linear range of the camera to prevent the overexposure of any blots.

## Radiochemistry

Clinical-grade [18F]FDG and [18F]fluoride were acquired from King's College London & Guys and St Thomas' PET Centre. [18F]FSPG was synthesised using a GE Fastlab according to previously published methodology[53].

## Radiotracer uptake and efflux

Cells were seeded into 6-well plates 24 h prior to uptake and efflux studies (see previous). For all cell uptake and efflux experiments, 0.185 MBq/mL of [18F]FSPG was added to wells (total volume 1 mL/well) and incubated for 1 h. Cells were maintained at 37 °C and 5% $CO_2$ throughout the uptake or efflux experiment. For uptake studies, plates were removed from the incubator after a 60 min incubation and placed on ice. Cells were washed three times with ice-cold PBS before the addition of 0.5 mL of RIPA buffer. 0.3 mL of cell lysate was taken for counting and the remaining 0.2 mL was used for protein quantification (BCA). For efflux experiments, following 60 min uptake, exogenous radioactivity was removed, with cells washed three times in room temperature PBS, and fresh media was added for 20, 40 or 60 min

before being placed on ice and processed as above. The amount of radioactivity in the samples was determined in a gamma counter (1282 Compugamma, LKB Wallac) and expressed as a percentage of the administered dose per mg protein.

## ROS quantification

ROS were detected in human NSCLC lines using the cell-permeable fluorophore CellROX Green (ThermoFisher Scientific, C10444). Cells were seeded in 6-well plates at the densities stated above in 2 mL media 24 h prior to performing the assay. A final concentration of 1 μmol/L CellROX Green reagent was added to each well and incubated for 30 min at 37 °C, protected from light. Cells were then washed with PBS, harvested using 0.05% trypsin-EDTA and suspended in 1 mL of ice-cold Hanks balanced salt solution (ThermoFisher Scientific, 14025092). The cell suspension was passed through a 35 μm filter and kept on ice prior to analysis on BD FACS Melody flow cytometer (488 nm laser and 527/32 bandpass filter; BD Biosciences). 20,000 single cell events were recorded per sample with the data gated post-acquisition based on forward (FS) and side scatter (SS) profiles to include only single cell events and to exclude cellular debris.

## Glutamate quantification

Intracellular glutamate concentrations were determined using a glutamate colorimetric assay kit, following the manufacturer's guidelines (Abcam, ab83389). Briefly, cells seeded in a 6-well plate were trypsinised, washed three times with ice-cold PBS and resuspended in 200 μL of assay buffer before sonication on ice (Soniprep 150, MSE). Lysates were then centrifuged at $15,000 \times g$ for 10 min at 4 °C and the supernatant taken for analysis. Total intracellular glutamate was normalised to protein concentration.

## GSH

Cells were seeded into 6-well plates as described above. Total GSH was determined using luminescence-based quantification (GSH/GSSG-Glo Assay, Promega, V6611). Cells were washed in ice-cold PBS, lysed in assay buffer and centrifuged at $15,000 \times g$ at 4 °C for 10 min. For in vivo studies, tumour tissue was dissected, snap-frozen in liquid nitrogen and stored at −80 °C. When required, tissue was placed into lysis Matrix tubes containing 1.4-mm ceramic beads (MP Biomedicals) and assay buffer (Promega). Tumour samples were lysed by rapid shaking using a high-speed benchtop reciprocating homogenizer (Fastprep-24 Sample Preparation Instrument; MP Biomedicals). The lysates were centrifuged at $15,000 \times g$ at 4 °C for 10 min, and the supernatant collected for analysis. For all studies, 5 μL of supernatant along with 5 μL of GSH standards (1–100 μmol/L) were added to white 96-well plates and total GSH was determined according to the manufacturer's instructions. Total intracellular GSH was normalised to protein concentration.

## Cystine

5 μL of conditioned media from cultured NSCLC cells subject to the indicated treatment was extracted in 195 μL of ice-cold 82% methanol supplemented with 25 mM N-ethylmaleimide, 10 mM ammonium formate and 10 μM [D4]cystine (Cambridge Isotope Laboratories, DLM-1000-1). Following a 30 min incubation at 4 °C, extracts were cleared by centrifugation at $17,000 \times g$ for 20 min at 4 °C. 5 μL of cleared extract was then analysed by LC-MS according to an established protocol[28]. Briefly, a Vanquish UPLC system was coupled to a Q Exactive HF (QE-HF) mass spectrometer equipped with HESI (Thermo Fisher Scientific) for chromatographic metabolite separation. Samples were then run on an Atlantis Premier BEH Z-HILIC VanGuard FIT column, 2.5 μm, 2.1 mm × 150 mm (Waters). The mobile phase A was 10 mM $(NH_4)_2CO_3$ and 0.05% $NH_4OH$ in $H_2O$, while mobile phase B was 100% ACN. The column chamber temperature was set to 30 °C. The mobile phase condition was set according to the following gradient:

0–13 min, 80–20% of mobile phase B; 13–15 min, 20% of mobile phase B. The ESI ionisation mode was negative, and the MS scan range (m/z) was set to 65–975. The mass resolution was 120,000, and the AGC target was $3 \times 10^6$. The capillary voltage and capillary temperature were set to 3.5 kV and 320 °C, respectively. The cystine and [D$_4$]cystine peaks were manually identified and integrated with EL-Maven (Version 0.11.0) by matching them to an in-house library. Cystine concentrations were calculated based on the 10 μM [D$_4$]cystine internal standard peak for each sample. Cystine consumption rates were calculated based on the change in media cystine levels following the 4 h culture duration and normalised to cell density.

### Analysis of transcriptomic data

Bulk RNA sequencing data were obtained from the TRACERx 421 study[31]. Once normal tissue and lymph node samples were removed, this comprised 882 samples from 344 NSCLC patients (median = 2 samples/patient; interquartile range: 2–3) with 279 having multi-region sampling. Z-score was calculated using the equation $z = (x-\mu)/\sigma$, where x is the raw score, μ is the population mean and σ is the population standard deviation.

### Orthotopic tumour cell implantation

$1 \times 10^6$ H460 FLuc cells or $5 \times 10^6$ H1299 FLuc were administered by a non-invasive intratracheal technique into the lungs of female NOD/SCID/IL2Rg$^{-/-}$ (NSG) mice aged 6–9 weeks (Charles River Laboratories)[21]. Briefly, mice were anaesthetised with isoflurane (2–2.5% in O$_2$) and transferred onto a vertical board, where they were suspended in an upright position by their upper incisors. Using a Leica M125 stereomicroscope (Leica Microsystems), the tongue was moved to one side to expose the vocal cords and the entrance to the trachea, where a plastic 20G i.v. catheter attached to a 1 mL syringe was inserted for cell administration, suspended in 50 μL of PBS. Tumour growth was monitored through bioluminescence imaging.

### Bioluminescence imaging

Orthotopic H460 FLuc and H1299 FLuc tumours were monitored through bioluminescence imaging using an IVIS Spectrum in vivo imaging system (PerkinElmer). Images were acquired 4 h post-cell inoculation to confirm the successful delivery of cells to the lung. Mice were subsequently imaged once a week for the first 3 weeks and then every 3–4 days thereafter. Mice were anaesthetised with isoflurane (1–2 % in O$_2$) and injected i.p. with 150 mg/kg firefly luciferin (Promega, P1041) before being transferred to the IVIS Spectrum camera and maintained at 37 °C. Images were acquired until the luminescent signal plateaued ~20 min p.i. of luciferin, ensuring maximum tumour signal was reached (exposure time 1–60 s, binning 2–8, FOV 23 cm, f/stop 1, no filter). Tumour growth was monitored until the experimental endpoint (mean tumour diameter of 1.5 cm). For signal quantification, images were analysed using Living Image software (PerkinElmer). A region of interest was drawn around the entire thorax and total photon flux was measured (photon/sec). Once the bioluminescent signal reached ~$5 \times 10^7$ photons/s/cm$^3$, mice were selected for PET/CT imaging.

### Genetically engineered mice

LSL-Kras$^{G12D/+}$ (RRID:IMSR_JAX:008179); p53$^{flox/flox}$ (RRID:IMSR_JAX:008462); LSL-Nrf2$^{D29H/+}$ (Nfe2l2tm1Gmdn, MGI: 7327101) mice[29] were housed and bred in accordance with the ethical regulations and approval of the University of South Florida Institutional Animal Care and Use Committee (protocol numbers: IS00005814M and IS00003893R). All mice were maintained on a mixed C57BL/6 genetic background. Lung tumours were induced by intranasal installation of $2.25 \times 10^7$ PFU adenoviral-Cre (University of Iowa)[29]. Adenoviral infections were performed under isofluorane anaesthesia. Following infection, mice were shipped to the U.K. for imaging studies.

### Patient-derived xenografts

Ethical approval to generate patient-derived models was obtained through the TRACERx clinical study (REC reference: 13/LO/1546; NCT01888601). PDX models were developed by subcutaneous injection of minced primary, surgically resected NSCLC tumour tissue in NSG mice[32]. PDX models were passaged twice before cryopreservation and shipping to King's College London for imaging experiments. Driver mutation profiles of PDX models were derived from whole-exome sequencing of both primary tumours and PDX models and are reported according to the definition previously described[32]. Cryopreserved tumour samples were thawed in a 37 °C water bath, transferred to a sterile 10 cm dish, and minced in fresh RPMI medium containing 1× penicillin/streptomycin. On ice, tumour pieces were then added to an Eppendorf containing 180 μL of Matrigel (Corning, 356234) before being injected into the upper flank of anesthetised female NSG mice (6–8 weeks) using a 19 G needle. Tumour growth was monitored using an electronic calliper, and the volume was calculated using the following equation: volume = $((\pi/6) \times h \times w \times l)$, where h, w and l represent height, width and length, respectively. Mice were monitored daily and selected for [$^{18}$F]FSPG PET/CT imaging when tumours reached ~150 mm$^3$.

### Longitudinal imaging of tumour xenografts

$3 \times 10^6$ H460 cancer cells in 100 μL PBS or $5 \times 10^6$ H1299 cancer cells in 50 μL PBS, 50 μL Matrigel were injected subcutaneously into female Balb/C nu/nu mice aged 6–9 weeks (Charles River Laboratories). Tumour growth was monitored by electronic callipers as described above. Mice were longitudinally imaged by [$^{18}$F]FSPG-PET across a range of tumour volumes, with thresholds set at 20–60 mm$^3$, 60–100 mm$^3$, 100–140 mm$^3$ and >140 mm$^3$.

### PET/CT imaging

Mice received a single bolus i.v. injection through a tail vein cannula of ~3 MBq [$^{18}$F]FSPG in 100 μL PBS. 40 min p.i., static PET imaging scans were acquired for 20 min on a Mediso NanoScan PET/CT system (1–5 coincidence mode; 3D reconstruction; CT attenuation-corrected; scatter corrected). CT images were acquired for anatomical visualisation and attenuation correction (720 projections; semi-circular acquisition; 55 kVp; 600 ms exposure time). Animals were maintained under anaesthesia (1–2% isoflurane in O$_2$) at 37 °C during tail vein cannulation, radiotracer administration and throughout the scan. Static image reconstruction was performed using the 3D Tera-Tomo algorithm, with 4 iterations, 6 subsets, a 0.4 mm isotropic voxel size and a binning window of 400–600 keV. VivoQuant software (v 2.5, Invicro Ltd) was used for image quantification of the reconstructed scans. Individual tumour volumes of interest in the lungs of mice were constructed by manually drawing sequential 2D regions of interest (ROI) on the CT images.

The percentage of tumour tissue in the lungs of individual mice was determined using the total CT voxel volume. Individual ROIs were manually drawn using VivoQuant from CT images, and the tumour volumes were summed for individual animals. Additionally, a single ROI was drawn around the entire lung to determine total lung volume. Together, the tumour volumes and total lung volume were used to calculate the percentage of tumour tissue present.

### Ex vivo tumour autoradiography

Following [$^{18}$F]FSPG PET/CT imaging, tumours were perfused with PBS prior to being embedded and frozen in a 1:1 OCT-PBS mixture using an isopentane bath over liquid nitrogen. The cryopreserved tumours were then sectioned (20 μm) using a Bright 5040 cryotome, with tissue sections thaw-mounted onto Superfrost PLUS glass microscope slides (Menzel-Glaser, Thermo). The slides were then exposed to a storage phosphor screen (Cytiva) and stored in a standard x-ray cassette overnight. The phosphor screen was then imaged using an Amersham

Typhoon scanner (Cytiva) and for anatomical reference, hematoxylin and eosin (H&E) staining was performed.

## Immunohistochemistry

Consecutive sections of formalin-fixed, paraffin-embedded tumour-containing lungs were used for immunohistochemical (IHC) analysis of xCT using a VECTOR DAB substrate kit (Vector laboratories) following the manufacturer's instructions. Slides were deparaffinized in xylene and rehydrated through graded ethanol to water before staining. For antigen unmasking, all sections were treated with 10 mM citrate solution (pH 6.0), and with 3% $H_2O_2$ to inactivate endogenous peroxidases. Sections were blocked in 2.5% normal blocking serum from the VECTASTAIN Elite ABC-HRP kit (Vector laboratories, PK-6100) prior to incubation with the avidin and biotin solutions, respectively (Vector laboratories). Sections were incubated with the anti-xCT antibody (1:200 dilution; AgilVax) overnight at 4 °C. After washing in TBST, the sections were stained with a biotinylated secondary antibody for 30 min at room temperature. After washing with TBST, sections were incubated with DAB substrate solution, and counterstained with haematoxylin. Images were acquired using a NanoZoomer (Hamamatsu), with representative images shown. Four areas were randomly selected from the tissue sections and the total number of cells and number of stained cells were counted. This was repeated for three individual tumours and the percentage of stained cells was calculated. For the genetically engineered mouse model studies, slides were scanned with the Aperio imager at 20× and the H-score of at least five representative regions/mouse was analysed with QuPath software[54]. Representative images were captured using the Axio Lab A1 microscope at 40× (Carl Zeiss Microimaging Inc.).

## HM30-tesirine growth inhibition in cells

H460 and H1299 cells were seeded into a 96-well plate at 2000 cells/well. An xCT-targeting mAb was linked to tesirine and provided by AgilVax (patent: WO2020/227640A1). 24 h post-seeding, cells were treated with increasing concentrations of HM30-tesirine ranging from 0.02 to 400 nM. After 72 h, media was removed and 100 μL of 0.5 mg/mL MTT solution was added for 4 h. Next, the MTT solution was removed, and formazan crystals were dissolved in 150 μL DMSO before absorbance was measured at 570 nm using a Varioskan Lux (Thermo-Fisher Scientific).

## HM30-tesirine treatment of NSCLC tumour-bearing mice

$3 \times 10^6$ H460 cancer cells in 100 μL PBS or $5 \times 10^6$ H1299 cancer cells in 50 μL PBS, 50 μL Matrigel (Corning) were injected subcutaneously into female Balb/C nu/nu mice aged 6–9 weeks (Charles River Laboratories). Tumour growth was monitored by electronic callipers as described above. When tumours reached ~70 mm³, mice were randomised into HM30-tesirine, cisplatin and vehicle treatment groups. HM30-tesirine-treated animals received three doses of 1.5 mg/kg on days 0, 7 and 14 via an i.p. injection (200 μL). The cisplatin-treated cohort received two doses of cisplatin (5 mg/kg; 200 μL i.p) on days 0 and 4. Vehicle control animals were treated on days 0 and 7. Mice were weighed and tumour volumes measured until humane endpoints were reached (mean tumour diameter of 1.5 cm).

## Statistics & reproducibility

Statistical analysis was performed using GraphPad Prism (v.8.0). All in vitro data was acquired from three or more independent replicates, acquired on separate days. Data were expressed as the mean ± one standard deviation (SD). Statistical significance was determined using unpaired two-tailed Student's t-test. For analysis across multiple samples, one-way analysis of variance (ANOVA) followed by t-tests multiple comparison correction (Tukey method) were performed.

Kaplan–Meier survival curve statistics were analysed with a log-rank (Mantel–Cox) test. To control the family-wise error rate in multiple comparisons, crude p-values were adjusted by the Holm–Bonferroni method. Differences with $p < 0.05$ were considered statistically significant in all analyses. No statistical method was used to predetermine the sample size. In Fig. 7e, tumour volumes for surviving mice were excluded when less than three mice remained in the treatment or control arm. No other data exclusions were performed. Mice for imaging and treatment studies were randomised into size-matched cohorts. Investigators were not blinded during data collection or analysis.

## Reporting summary

Further information on research design is available in the Nature Portfolio Reporting Summary linked to this article.

## Data availability

All data are available in the article and its Supplementary materials. The DNA sequencing data from patients and PDXs used in this study has been deposited in the European Genome-Phenome Archive under accession code EGAS00001006494. The corresponding RNA sequencing data is available in the same database under the accession code EGAS00001006494. Access to DNA and RNA sequencing data is restricted to comply with patient consent and to determine the scientific purpose of data reuse. The Francis Crick Institute, which owns the data, will provide access to published datasets upon written application and signature of a Data Access Agreement. Applications require approval by the TRACERx Data Access Committee, who will provide the data access form upon contact. Source data are provided with this paper.

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

## Acknowledgements

The authors thank members of the lung TRACERx consortium, in particular Mariam Jamal-Hanjani and Crispin Hiley, as well as the patients who enroled in the study. We are grateful for the support provided by Tammy Kalber and Stephen Patrick, who facilitated the GEMM imaging studies at UCL, and both Kavitha Sunassee and Jana Kim, who supported the imaging studies at KCL. Finally, we thank Les Stewart and Joseph Patti from AgilVax for insightful discussions and for providing HM30-tesirine. This study was funded through a Wellcome Trust Senior Research Fellowship (220221/Z/20/Z) and by UK Research and Innovation (UKRI) under the UK government's Horizon Europe funding guarantee (grant number 10091247) to T.H.W., arising from the European Innovation Council Pathfinder grant (Grant No. 101129886,

SMARTdrugs). G.M.D received funding from the NIH/NCI (R37-CA230042). R.E.H. was funded by a Wellcome Trust Sir Henry Wellcome Fellowship (WT209199/Z/17/Z) and received additional support through the Cancer Research UK Lung Cancer Centre of Excellence. This work was supported by the Francis Crick Institute, which receives its core funding from Cancer Research UK, the UK Medical Research Council, and the Wellcome Trust (CC2041).

## Author contributions

H.E.G designed and performed the experiments, developed the methodology, analysed the data, formatted the figures, and wrote the manuscript. A.R.B performed the in vivo efficacy studies and analysed the resulting data. R.S.E performed [$^{18}$F]FSPG radiosynthesis and contributed to study design and data interpretation. W.E.T, M.E.G, S.N.S, G.F., L.M.S. and O.V.T. performed in vivo imaging studies and analysed the data. F.B. developed the orthotopic NSCLC animal model. M.T. performed [$^{18}$F]FSPG radiosynthesis. E.K. contributed towards the design of the work and performed preliminary data acquisition. A.F. maintained and genotyped the KP/KPN mouse colony and infected mice with adenoviral Cre to generate tumour-bearing mice. N.P.W. performed mass spec analysis of cystine consumption. J.M.D performed immunohistochemistry analysis of xCT expression and its association with tumour grade in the KP and KPN GEMM model. L.T. generated the NRF2 KO/KOR A549 and NRF2 T80K expressing H1299 cell systems. P.N.S performed western blotting for NRF2 and xCT expression in the NRF2 KO/KOR A549 and NRF2 T80K expressing H1299 cell systems. A.H. analysed TRACERx patient RNA transcriptomic data and NSCLC PDX whole-exome sequencing data. D.R.P. and R.E.H. initiated and expanded NSCLC PDX models, and provided training in their use. C.S. and R.E.H. provided NSCLC PDX models. R.E.H. edited a draft and revised the manuscript. G.M.D. developed the KPN GEMM, provided cell lines and resources, metabolomic expertise, edited a draft and revised the manuscript and contributed to the experimental design. T.H.W. conceived the study, acquired funding, contributed to the experimental design, supervised the project, designed the figures and wrote the manuscript. All authors approved the submitted version of the manuscript.

## Competing interests

The authors declare no competing interests.
