## [Peer Review File · Nature Communications]

REVIEWER COMMENTS

Reviewer #1 (Cancer imaging/PET):

In the manuscript by Greenwood et al, the authors examined whether tumor-intrinsic KEAP1/NRF2 mutations could be non-invasively imaged by [¹⁸F]FSPG PET imaging and whether aberrant system xc⁻ expression could be exploited for the treatment of refractive NSCLC tumors. As a conclusion, the authors claim that they have comprehensively demonstrated the ability of system xc⁻ to report on NRF2 activation using a combination of metabolomics, genetic engineering, drug treatment, and biochemical analyses. The manuscript is well written, allowing readers to easily follow along, and each experiment's logic and direction are well-designed. I appreciate the extensive experiments using cell lines, PDXs, genetically engineered mouse models, and patient bulk RNA-seq data, and there are several interesting results in the study. However, to enhance the completeness of this research, the reviewer would like to leave the following comments that I feel should be addressed to improve the manuscript.

1. Do you believe that the results presented in Figure 2 i and j demonstrate differing outcomes between the A549 and H1299 cell lines due to the H1299's low baseline NRF2 protein expression? Additionally, similar to the treatment with KI696, I'm curious if you have also verified changes in NRF2 protein expression resulting from this genetic manipulation.

2. In Extended Figure 2, you present the results of bioluminescence imaging (BLI) from mice orthotopically modeled with H460 (NRF high) and H1299 (NRF low) cell lines. Was a quantitative assessment conducted? It's challenging to discern differences based on the images alone; could you provide further explanation as to why these results were obtained?

3. When examining the results from all mouse PET images, it appears that analyses were conducted using multiple lesions per lung per mouse. How were you able to identify and analyze each lesion individually in the small lungs of mice? It's possible that errors could occur in the analysis. Additionally, how were comparisons made in the statistical analysis? Using the uptake values from all lesions across the two cell line groups might not be the accurate approach. For example, if Mouse 1 has 4 lesions and Mouse 2 has 3 lesions, calculating a mean or median to represent each mouse seems necessary, and then those representative values should be used to conduct statistical comparisons between the two groups.

4. Could you please clearly describe the criteria used for the selection of PDXs in Extended Figure 5?

5. In Figures 6g and h, it would be beneficial to include further discussion on the results showing no difference in Cleaved Caspase 3 staining between the control group and HM30-tesirine treatment group.

6. As minor comments, I would like to mention the following: In the scheme of Figure 6d, please consider changing to nude mice to avoid the impression, based solely on the figure, that mice with fur, such as NSG mice, were used. For Figures 6e and f, it might be beneficial to change the legend's name to "ADC" to "HM30-tesirine," similar to what was done in Extended Data Figure 8.

7. I think the title is overstated and too broad, the implication is that NRF2 is directly imaged; yet xct activity is a downstream effect that is coupled. The authors might consider a title that more directly associates NRF2 and XCT/FSPG PET uptake.

Reviewer #2 (Cancer metabolism):

Summary: Greenwood et al. utilized positron emission tomography (PET) and demonstrated its efficacy as a non-invasive marker for NRF2 activation in advanced NSCLC models, which was validated by genetic and pharmacological manipulation. Importantly, they confirmed the correlation between [18F]FSPG PET imaging and system xc⁻ activity by metabolomic analyses, as well as intracellular glutathione concentration. The authors advocate for the clinical evaluation of [18F]FSPG PET as a predictive marker for therapy resistance in NSCLC, highlighting the potential for personalized treatment strategies targeting the NRF2-KEAP1 pathway.

The basic findings of this study are very exciting to the field of cancer biology and cancer metabolism and clinical imaging. The molecular basis of the mechanism behind xc⁻ activity and [18F]FSPG PET imaging signal is well established (PMID: 29471880, 35191738, 32694158), however this study provides a link between the already known role of NRF2 in regulating xCT and an important technical advancement. In order to improve their manuscript, the authors should address the following major concerns.

Revisions:

1. The authors should test HM30-tesirine on H1975, H1299, H23 and H1650 with and without KI696 as their xCT levels are affected by KI696 (fig 2b). That would significantly strengthen their claim of differential efficacy of targeting xCT high cells.
2. The authors claim that “[18F]FSPG provides a sensitive and specific marker of NRF2 activation in preclinical models”. To be considered a marker and establish its specificity and sensitivity, the authors should perform following studies: Longitudinal (early, midtime and late) PET imaging of in vivo tumors with different status of NRF2 pathway activation. Establishing at which stages PET imaging alone is enough to discriminate status of NRF2 activation. The results from figure 4i it suggest that there might be specific stages of tumor development that provide better sensitivity to distinguish NRF2^{high} vs NRF2^{low} tumors. The question is can proposed PET imaging differentiate NRF2 status among tumors of A) same time point after tumor initiation B) Different time point after tumor initiation – reflecting different tumor stages in clinical settings. The mathematical models to establish if some readout can be considered as a marker are well established - rROC curve, false positivity rate, false negative rate, true positive and true negative rates. The calculation for all of those should be incorporated into the manuscript if the authors want to consider [18F]FSPG as a marker.
3. The HM30 treatment in vivo started 7-10 days post tumor implantation (Fig 6d). However, there is not discussion whether at that stage of tumor size/development [18F]FSPG can distinguish with enough sensitivity the NRF2 status of those tumors. The authors need to evaluate that in order to claim that FSPG imaging has any usefulness in clinical settings.
4. The Experiment in figure 6d has been performed with NRF2 high H460 model. In those settings, HM30 suppressed tumor growth of xc- high tumors. However, HM30 has not been tested in vivo in the NRF2 low settings (for example H1299). That experiment is necessary in order to back up 2 crucial claims of the manuscript: 1) That FSPG imaging can be beneficial strategy in clinical settings which is tightly connected with 2) We have treatment options that will significantly delay tumor progression of specific subset of tumors we can differentiate (stratify) by our imaging tool. These are two major conclusions of this study that need to be addressed experimentally. The PET imaging has to be able to differentiate the NRF2 status of tumors before the treatment is initiated, and secondly, the treatment should be significantly more beneficial with tumors of high NRF2 status (in vivo with HM30 and NRF2 low tumors is necessary for that to be established).

Reviewer #3 (Lung cancer therapy, preclinical model):

This is a very interesting study that addresses the visualization and therapeutic targeting of NRF2 activation in lung tumors using PET-CT imaging. KEAP1 inactivation causes NRF2 activation, which is associated with resistance to current treatments. The study shows that cell lines with elevated NRF2 exhibit increased xCT and cysteine consumption, GSH production, and low glutamate levels.

The uptake of the xCT substrate [¹⁸F]FSPG in cells correlates with GSH levels. NRF2 inhibition by an inhibitor reduced cysteine consumption and GSH levels.

The authors visualized orthotopic lung tumors with human cell lines using [¹⁸F]FSPG PET. Mouse tumors with an NRF2 activating mutation in the background of KP can also be visualized using this method. These tumors have high xCT expression. The authors identified an antioxidant signature in a subset of PDXs and used a pair of high and low signatures for imaging with [¹⁸F]FSPG. They then tested the efficacy of a humanized xCT antibody in the H460 cell line in vivo.

This study describes a novel imaging and treatment modality. It highlights a new tumor imaging marker and a new therapeutic strategy for these tumors.

Comments:

1. The association of xCT expression with KEAP1 mutations needs to be addressed. Can the authors show this across all human lung cancer cells? Is this connection lung cancer-specific?
2. The association of KEAP1 mutations with xCT needs to be shown using isogenic cell line pairs; restoration of KEAP1 will be needed to confirm this is not cell line dependent.
3. How does KI696 work, and is it specific to KEAP1?
4. The genetic makeup of the cells used in the experiments needs to be clarified. For example, does the STK11 status also change GSH accumulation and FSPG uptake?
5. How sensitive is the visualization? Longitudinal follow-up of the signal and tumor quantification by CT needs to be graphed for the models shown in the main figures.
6. The signal from the GI seems different in each mouse and figure. Can the authors normalize the background signal or exposure? The signal in normal organs is quite high. How do the authors propose visualization for metastatic tumors? This should be stated as a limitation.
7. How does FDG glucose imaging look in these pairs of tumors? Is this a general metabolic uptake or specific to FSPG?
8. Imaging of the KP tumor is impressive, and the authors nicely complement this homogeneous model with the more heterogeneous PDX. What do the authors think about NRF2 activation in these mice versus actual tumors? How do the levels correlate?
9. In figure 4d, xCT expression seems not on the membrane. How is the ADC working? Also, how will the toxicity in normal tissues be addressed?
10. In figure 5a the expression of the NRF2 gene signature appears low in most samples. Was that the conclusion from this data?
11. Can this system detect small PDX tumors?
12. Does the xCT-ADC work on NRF2-low tumors?

Reviewers' Comments:

Reviewer #1 (Cancer imaging/PET)

In the manuscript by Greenwood et al, the authors examined whether tumor-intrinsic KEAP1/NRF2 mutations could be non-invasively imaged by [¹⁸F]FSPG PET imaging and whether aberrant system xc⁻ expression could be exploited for the treatment of refractive NSCLC tumors. As a conclusion, the authors claim that they have comprehensively demonstrated the ability of system xc⁻ to report on NRF2 activation using a combination of metabolomics, genetic engineering, drug treatment, and biochemical analyses. The manuscript is well written, allowing readers to easily follow along, and each experiment's logic and direction are well-designed. I appreciate the extensive experiments using cell lines, PDXs, genetically engineered mouse models, and patient bulk RNA-seq data, and there are several interesting results in the study. However, to enhance the completeness of this research, the reviewer would like to leave the following comments that I feel should be addressed to improve the manuscript.

We are grateful for the reviewer's positive comments and for their carefully considered review.

1. Do you believe that the results presented in Figure 2 i and j demonstrate differing outcomes between the A549 and H1299 cell lines due to the H1299's low baseline NRF2 protein expression? Additionally, similar to the treatment with KI696, I'm curious if you have also verified changes in NRF2 protein expression resulting from this genetic manipulation.

The differences in terms of the magnitude of changes in [¹⁸F]FSPG following genetic manipulation in A549 and H1299 is almost certainly related to low baseline xCT expression in H1299 cells. Even after ectopic expression of NRF2^{T80K}, xCT protein expression also doesn't reach the levels seen in A549 cells. We have included a western blot showing NRF2 and xCT expression side-by-side in the isogenic lines (**Fig. 2g**) in our updated Figure 2, shown below. We have also updated the associated manuscript text as follows:

"In agreement with our previously published work²⁴, knockdown (KD) of NRF2 in A549 cells reduced xCT expression, which was restored through ectopic expression of NRF2 (KD-R). Introducing the NRF2^{T80K} mutation to H1299 cells increased NRF2 expression in H1299 cells compared to empty-vector controls (PLX317), which resulted in a less-pronounced increase in xCT."

Figure 2. $[^{18}\text{F}]$ FSPG retention is altered following pharmacological and genetic manipulation of NRF2. *a*, Chemical structure of KI696. *b*, Representative western blot of NRF2 and xCT expression in NRF2-low cell lines 24 h post treatment with vehicle control or 200 μM KI696. Actin was used as a loading control. *c-f*, Analysis of cystine (Cys_2) consumption (*c*), intracellular glutamate (*d*) and intracellular GSH (*e*) in NRF2-low lines following KI696 treatment compared to vehicle control. *f*, Intracellular $[^{18}\text{F}]$ FSPG retention in NRF2-low cells after KI696 treatment compared to vehicle control. *g*, Representative western blot of NRF2 and xCT expression in NSCLC cells following genetic manipulation of NRF2. *h-k*, Intracellular GSH (*h,j*) and $[^{18}\text{F}]$ FSPG retention (*i,k*) in genetically modified NSCLC cells. Data are presented as mean \pm SD. *, $p < 0.05$; **, $p < 0.01$; ***, $p < 0.001$.

2. In Extended Figure 2, you present the results of bioluminescence imaging (BLI) from mice orthotopically modeled with H460 (NRF high) and H1299 (NRF low) cell lines. Was a quantitative assessment conducted? It's challenging to discern differences based on the images alone; could you provide further explanation as to why these results were obtained?

We used bioluminescence to qualitatively assess tumour engraftment in the lungs soon after injection and longitudinally monitor total tumour burden. This acted as a pre-screen to select tumour-bearing animals before $[^{18}\text{F}]$ FSPG imaging. Due to the scattering of light and variable tumour tissue depth, these images are semi-

quantitative at best. By thresholding the signal at the same scale, however, tumour growth can be observed, as shown in Extended Data Fig. 2.

We have amended the text as follows to provide further clarity:

“Initial tumour engraftment and qualitative assessment of tumour growth was monitored by bioluminescence imaging (BLI; **Extended Data Fig. 2**)...”

3. When examining the results from all mouse PET images, it appears that analyses were conducted using multiple lesions per lung per mouse. How were you able to identify and analyze each lesion individually in the small lungs of mice? It's possible that errors could occur in the analysis.

You are correct that all PET analyses were performed on a lesion-by-lesion basis. As shown in Figure 4b, the high-resolution CT scans used in this study, combined with excellent tumour-to-lung contrast, permitted accurate delineation of small individual tumours within the lung. Whilst micrometastases are not visible, using CT as the ground-truth and deriving quantitative data from the overlaid PET images permits a high degree of accuracy.

Additionally, how were comparisons made in the statistical analysis? Using the uptake values from all lesions across the two cell line groups might not be the accurate approach.

For example, if Mouse 1 has 4 lesions and Mouse 2 has 3 lesions, calculating a mean or median to represent each mouse seems necessary, and then those representative values should be used to conduct statistical comparisons between the two groups.

In our analysis, we compared all lesions across both tumour types. By including all lesions in this manner, we can obtain further experimental data related to interlesion heterogeneity that would otherwise be lost if each mouse was reduced to a simple averaged data point. A good example of why we are keen to include all lesions in our analysis is in Fig. 4e, which highlights the large spread of [¹⁸F]FSPG tumour uptake in KPN mice, with some lesions reaching >80% ID/g. Consequently, we think it is appropriate to keep the statistical analysis as it is currently presented. Our work is still sufficiently powered in terms of individual mice used, and we have provided the numbers of animals and lesions directly below the quantification to maintain transparency.

4. Could you please clearly describe the criteria used for the selection of PDXs in Extended Figure 5?

The PDXs used in Fig.5 and extended Fig. 5 were randomly selected from PDXs known to grow *in vivo* that were either *NFE2L2* wildtype or contained truncal alterations in *NFE2L2*. This selection criteria is described in the text:

“We identified patient-derived xenograft (PDX) models from TRACERx patients which were either *NFE2L2* wildtype (WT; CRUK0640 region 8; R8) or contained truncal alterations in *NFE2L2* (c.G187C/p.E63Q and c.G352C/p.E118Q; CRUK0772

R1)³⁴.”

5. In Figures 6g and h, it would be beneficial to include further discussion on the results showing no difference in Cleaved Caspase 3 staining between the control group and HM30-tesirine treatment group.

Immunohistochemical analyses were performed at endpoint, long after the mice had been off treatment. Presumably, by this time, the innate immune system had already removed all dead cells. We have updated the text as follows (new text is underlined):

“At these late time points, there was no difference in levels of apoptosis between groups, as shown by cleaved caspase 3 staining ($32 \pm 11\%$ positive cells for control vs. $28 \pm 5.7\%$ for HM30-tesirine; $n = 3$; $p = 0.66$; **Fig. 7g,h**), presumably due to the removal of dead cells by the innate immune system.”

6. As minor comments, I would like to mention the following: In the scheme of Figure 6d, please consider changing to nude mice to avoid the impression, based solely on the figure, that mice with fur, such as NSG mice, were used.

Thank you for identifying this error. We have removed this schematic in the revised manuscript, replacing it with newly-acquired imaging data.

For Figures 6e and f, it might be beneficial to change the legend's name to "ADC" to "HM30-tesirine," similar to what was done in Extended Data Figure 8.

This has been changed in the revised manuscript.

7. I think the title is overstated and too broad, the implication is that NRF2 is directly imaged; yet xct activity is a downstream effect that is coupled. The authors might consider a title that more directly associates NRF2 and XCT/FSPG PET uptake.

We agree and have changed the title to: “Imaging NRF2 activation in non-small cell lung cancer with positron emission tomography”

Reviewer #2 (Cancer metabolism)

Summary: Greenwood et al. utilized positron emission tomography (PET) and demonstrated its efficacy as a non-invasive marker for NRF2 activation in advanced NSCLC models, which was validated by genetic and pharmacological manipulation. Importantly, they confirmed the correlation between [18F]FSPG PET imaging and system xc⁻ activity by metabolomic analyses, as well as intracellular glutathione concentration. The authors advocate for the clinical evaluation of [18F]FSPG PET as a predictive marker for therapy resistance in NSCLC, highlighting the potential for personalized treatment strategies targeting the NRF2-KEAP1 pathway.

The basic findings of this study are very exciting to the field of cancer biology and cancer metabolism and clinical imaging. The molecular basis of the mechanism behind xc⁻ activity and [18F]FSPG PET imaging signal is well established (PMID: 29471880, 35191738, 32694158), however this study provides a link between the already known role of NRF2 in regulating xCT and an important technical advancement. In order to improve their manuscript, the authors should address the following major concerns.

Thank you for your generous comments, highlighting the clinical importance of these findings.

Revisions:

1. The authors should test HM30-tesirine on H1975, H1299, H23 and H1650 with and without KI696 as their xCT levels are affected by KI696 (fig 2b). That would significantly strengthen their claim of differential efficacy of targeting xCT high cells.

We agree that the experiment described above would have provided important information related to the differential targeting of xCT-high vs xCT-low cells. Unfortunately, when we performed this experiment, 72h KI696 treatment alone induced extensive cell death. 72h is required for HM30-tesirine-mediated cell killing in cell culture and in the data presented in Fig.2 was only used for 24h; a time when no cell death was evident. Given the differential in treatment efficacy seen *in vivo* between NRF2-high and NRF2-low (see below), this negates the need for further imperfect *in vitro* investigation.

2. The authors claim that “[18F]FSPG provides a sensitive and specific marker of NRF2 activation in preclinical models”. To be considered a marker and establish its specificity and sensitivity, the authors should perform following studies: Longitudinal (early, midtime and late) PET imaging of *in vivo* tumors with different status of NRF2 pathway activation. Establishing at which stages PET imaging alone is enough to discriminate status of NRF2 activation. The results from figure 4i it suggest that there might be specific stages of tumor development that provide better sensitivity to distinguish NRF2high vs NRF2low tumors. The question is can proposed PET imaging differentiate NRF2 status among tumors of A) same time point after tumor initiation B) Different time point after tumor initiation – reflecting different tumor stages in clinical settings.

We performed new longitudinal [¹⁸F]FSPG imaging in NRF2-high H460 and NRF2-low H1299 subcutaneous tumours to address this important point. As the growth rates of these two tumours are quite different, we chose to image mice not as a function of time post-initiation but as a function of tumour size, starting off with very small volumes of 20-60 mm³. Even in these small tumours, there was a statistically significant difference in [¹⁸F]FSPG tumour retention between the two tumour types; a finding that was reproduced across all tumour sizes measured. At volumes >140 mm³ H460 tumours became necrotic, thereby reducing the average [¹⁸F]FSPG signal retained over the entire tumour volume. Clinically, simple SUV measurements of the single most intense voxel of a given volume will reduce issues related to average measurements. However, even in the >140 mm³ cohort, there was still a clear differential between NRF2-high and NRF2-low tumours.

Taken together, we believe these data further strengthen the evidence that PET imaging can differentiate NRF2 pathway activation in tumours. In-turn, this provides a non-invasive biomarker for treatment stratification.

The new data (Fig. 7d and Extended Figure 8) and associated text can be found below:

Figure 7. HM30-tesirine controls tumour growth and prolongs survival of mice bearing H460 tumours. *a*, Structure of the anti-xCT tesirine conjugate, HM30-tesirine. *b*, Western blot using HM30-tesirine as the primary antibody in H1299 and

H460 cell lysates. Actin was used as a loading control. c, HM30-tesirine MTT dose-response in H460 and H1299 cells. d, Axial [^{18}F]FSPG PET/CT images from mice bearing H460 or H1299 tumours before initiation of treatment. The dashed circle indicates the tumour. e-f, Antitumour activity (e) and survival benefit (f) of control (saline treated), cisplatin treated, and HM30-tesirine treated mice. The arrows under the x-axis of (e) represent treatment cycles for cisplatin (green) and HM30-tesirine (red). g, IHC for Ki67 and cleaved caspase 3 from FFPE tumours taken at endpoint. Scale bar, 50 μm . h-i, Corresponding quantification of tissue staining for Ki67 (h) and cleaved caspase 3 (i). Data are presented as the mean values \pm SD ($n = 3$). **, $p < 0.01$; ***, $p < 0.001$.

Extended Figure 8. [^{18}F]FSPG can distinguish NRF2-high from NRF2-low-expressing tumours across a range of tumour volumes. a, Representative axial [^{18}F]FSPG PET/CT images of 40-60 min summed activity in mice bearing subcutaneous H460 (NRF2-high) and H1299 (NRF2-low) tumours of varying size. Dashed white lines indicate the tumour. b, Quantification of [^{18}F]FSPG tumour retention. *, $p < 0.05$; **, $p < 0.01$; ***, $p < 0.001$.

Materials and methods:

“Longitudinal imaging of tumour xenografts

3×10^6 H460 cancer cells in 100 μL PBS or 5×10^6 H1299 cancer cells in 50 μL PBS, 50 μL Matrigel (Corning) were injected subcutaneously into female Balb/C

nu/nu mice aged 6-9 weeks (Charles River Laboratories). Tumour growth was monitored using an electronic calliper, and the volume was calculated using the following equation: $\text{volume} = ((\pi/6) \times h \times w \times l)$, where h, w and l represent height, width, and length, respectively. Mice were longitudinally imaged by [¹⁸F]FSPG-PET across a range of tumour volumes, with thresholds set at 20-60 mm³, 60-100 mm³, 100-140 mm³ and >140 mm³.

Results:

“We have shown above that [¹⁸F]FSPG measures NRF2 pathway activation with high sensitivity and specificity. Next, we performed longitudinal PET imaging in mice bearing subcutaneous H460 or H1299 tumours with the aim of stratifying responders from non-responders. There was a marked and statistically significant difference in [¹⁸F]FSPG retention between these two tumour types (**Fig. 7d**). This pattern was replicated across all tumour sizes and stages of development, establishing that even in very small tumours (20 – 60 mm³), [¹⁸F]FSPG can discriminate NRF2 activation status (**Extended Fig. 8**). [¹⁸F]FDG, however, was unable to differentiate between NRF2-high and NRF2-low tumours (**Extended Fig. 9**).”

The mathematical models to establish if some readout can be considered as a marker are well established - rROC curve, false positivity rate, false negative rate, true positive and true negative rates. The calculation for all of those should be incorporated into the manuscript if the authors want to consider [¹⁸F]FSPG as a marker.

We completely agree and have added ROC curves for orthotopic, genetically engineered, and patient-derived tumours. The results are remarkable, with each tumour model approaching an area under the receiver-operating characteristic curve of 1. We have included a new section in the revised manuscript and a new accompanying Figure:

“¹⁸F]FSPG identifies NRF2 pathway activation with high sensitivity and specificity

To better understand the degree by which [¹⁸F]FSPG can distinguish NRF2-high from NRF2-low tumours, we analysed the areas under the receiver-operating characteristic (ROC) curve for each *in vivo* model. The ROC illustrates the performance of a binary classifier model at varying threshold levels, with the true positive rate (y-axis) plotted against the false positive rate (x-axis) at each threshold. The area under the curve (AUC) provides the overall performance of [¹⁸F]FSPG as a binary classifier across all thresholds. For all tumour models evaluated in this study (**Fig. 3, 4 and 5**), [¹⁸F]FSPG had AUC values approaching 1, at 0.973 ($p = 0.0002$), 0.996 ($p < 0.0001$) and 0.990 ($p = 0.0002$) for orthotopic, GEMM, and PDX tumours, respectively (**Fig. 6**).

Figure 6. [^{18}F]FSPG can differentiate NRF2 activation status with high sensitivity and specificity. ROC analysis of PET imaging data in orthotopic (a), GEMM (b), and PDX (c) tumours.

3. The HM30 treatment in vivo started 7-10 days post tumor implantation (Fig 6d). However, there is not discussion whether at that stage of tumor size/development [^{18}F]FSPG can distinguish with enough sensitivity the NRF2 status of those tumors. The authors need to evaluate that in order to claim that FSPG imaging has any usefulness in clinical settings.

We treated tumours at 70 mm^3 , at which point they were well-vascularised. As shown in point 2, there was a clear differential between NRF2-high and NRF2-low tumours at this volume, shown by [^{18}F]FSPG-PET. We have clarified this in the amended text, which is preceded by the paragraph describing the ability of FSPG to differentiate NRF2-high from -low tumours with high sensitivity:

“Having stratified tumours as [^{18}F]FSPG-high (H460) or [^{18}F]FSPG-low (H1299), the *in vivo* efficacy of HM30-tesirine was examined in tumour-bearing mice. Tumours were grown subcutaneously until they reached $\sim 70\text{ mm}^3$ – a point where they were well-vascularised and had differential [^{18}F]FSPG retention”

4. The Experiment in figure 6d has been performed with NRF2 high H460 model. In those settings, HM30 suppressed tumor growth of xc- high tumors. However, HM30 has not been tested in vivo in the NRF2 low settings (for example H1299). That experiment is necessary in order to back up 2 crucial claims of the manuscript: 1) That FSPG imaging can be beneficial strategy in clinical settings which is tightly connected with 2) We have treatment options that will significantly delay tumor progression of specific subset of tumors we can differentiate (stratify) by our imaging tool. These are two major conclusions of this study that need to be addressed experimentally. The PET imaging has to be able to differentiate the NRF2 status of tumors before the treatment is initiated, and secondly, the treatment should be significantly more beneficial with tumors of high NRF2 status (in vivo with HM30 and NRF2 low tumors is necessary for that to be established).

Again, we are in complete agreement with the importance of these experiments. In our original manuscript, the assumption was that only xCT-high tumours

(distinguishable by imaging) would benefit from HM30-tesirine. We did not experimentally prove that negative tumours also would not benefit.

In the revised manuscript, we performed new HM30-tesirine efficacy studies in the H1299 tumour model (NRF2-low). As would have been predicted both from the underlying biology and the FSPG imaging studies, HM30-tesirine had no efficacy in this model. Taken together, FSPG can, therefore, differentiate NRF2-high from NRF2-low tumours with high sensitivity and specificity, which enables the stratification of subjects into responders and non-responders.

We have produced a new extended Figure that contains this new data and have adjusted the text accordingly (new text is underlined):

Extended Figure 11. HM30-tesirine has no efficacy in NRF2-low H1299 tumours. Anti-tumour activity (a), survival benefit (b) and animal weights (c) of control (saline treated), cisplatin treated, and HM30-tesirine treated mice. The arrows under the x-axis of a and c represent treatment cycles for cisplatin (green) and HM30-tesirine (red).

Results (p28):

“Conversely, HM30-tesirine did not affect tumour growth or survival in NRF2- and xCT-low H1299 tumours compared to vehicle control animals, whereas cisplatin induced a growth delay (**Extended Data Fig. 11**).”

Discussion (p32):

“Importantly, HM30-tesirine significantly extended the lives of H460 tumour-bearing mice compared to both cisplatin- and vehicle-treated animals. HM30-tesirine was ineffective in NRF2 WT tumours, highlighting the importance of [¹⁸F]FSPG in stratifying responders from non-responders.”

We hope that through these additional new in vivo experiments we have made a convincing case that FSPG imaging is a beneficial strategy to select subjects that are likely to benefit from our new xCT-targeted therapeutic.

Reviewer #3 (Lung cancer therapy, preclinical model)

This is a very interesting study that addresses the visualization and therapeutic targeting of NRF2 activation in lung tumors using PET-CT imaging. KEAP1 inactivation causes NRF2 activation, which is associated with resistance to current treatments. The study shows that cell lines with elevated NRF2 exhibit increased xCT and cysteine consumption, GSH production, and low glutamate levels. The uptake of the xCT substrate [¹⁸F]FSPG in cells correlates with GSH levels. NRF2 inhibition by an inhibitor reduced cysteine consumption and GSH levels.

The authors visualized orthotopic lung tumors with human cell lines using [¹⁸F]FSPG PET. Mouse tumors with an NRF2 activating mutation in the background of KP can also be visualized using this method. These tumors have high xCT expression. The authors identified an antioxidant signature in a subset of PDXs and used a pair of high and low signatures for imaging with [¹⁸F]FSPG. They then tested the efficacy of a humanized xCT antibody in the H460 cell line in vivo. This study describes a novel imaging and treatment modality. It highlights a new tumor imaging marker and a new therapeutic strategy for these tumors.

Thank you for your thorough description of our work, highlighting key advances in the field.

Comments:

1. The association of xCT expression with KEAP1 mutations needs to be addressed. Can the authors show this across all human lung cancer cells? Is this connection lung cancer-specific?

This is an important point that is addressed in Fig.1b. Here, we show the tight association between KEAP1 mutations, NRF2-, and xCT-expression. This has been reproduced here for your reference.

KEAP1/NRF2 activating mutations are present in approximately 1/3rd of all NSCLC patients (Cancer Genome Atlas Research. Comprehensive molecular profiling of lung adenocarcinoma. *Nature* 511, 543-550 (2014)). NRF2 is overexpressed in a range of tumour types of different histological origin, particularly in response to therapy-induced oxidative stress. Pancreatic ductal adenocarcinoma, in particular, is a promising indication as NRF2 is overexpressed in nearly all tumour resection samples analysed to date (Hayes et al. Keap1–Nrf2 signalling in pancreatic cancer. *Int J Biochem Cell Biol.* 65, 288-299 (2015)). This tumour type will be the focus of future work from my Group.

2. The association of KEAP1 mutations with xCT needs to be shown using isogenic cell line pairs; restoration of KEAP1 will be needed to confirm this is not cell line dependent.

In the isogenic cell experiments, we have functionally knocked out NRF2 and restored ectopic expression in A549 cells, whilst introducing a NRF2 T80K mutation in NRF2 wild-type H1299 cells. Altering NRF2 activity is interchangeable with KEAP1 modulation. We apologise for omitting the westerns validating our isogenic models. They have been included in the revised manuscript as a new figure part (**Fig. 2g**) in our updated Figure 2, shown below. We have also updated the associated manuscript text as follows:

“In agreement with our previously published work²⁴, knockdown (KD) of NRF2 in A549 cells reduced xCT expression, which was restored through ectopic expression of NRF2 (KD-R). Introducing the NRF2^{T80K} mutation to H1299 cells increased NRF2 expression in H1299 cells compared to empty-vector controls (PLX317), which resulted in a less-pronounced increase in xCT.”

Figure 2. [¹⁸F]FSPG retention is altered following pharmacological and genetic manipulation of NRF2. *a*, Chemical structure of KI696. *b*, Representative western blot of NRF2 and xCT expression in NRF2-low cell lines 24 h post treatment with vehicle control or 200 μM KI696. Actin was used as a loading control. *c-f*, Analysis of cystine (Cys₂) consumption (*c*), intracellular glutamate (*d*) and intracellular GSH (*e*) in NRF2-low lines following KI696 treatment compared to vehicle control. *f*, Intracellular [¹⁸F]FSPG retention in NRF2-low cells after KI696 treatment compared to vehicle control. *g*, Representative western blot of NRF2 and xCT expression in NSCLC cells following genetic manipulation of NRF2. *h-k*, Intracellular GSH (*h,j*) and [¹⁸F]FSPG retention (*i,k*) in genetically modified NSCLC cells. Data are presented as mean ± SD. *, *p* < 0.05; **, *p* < 0.01; ***, *p* < 0.001.

3. How does KI696 work, and is it specific to KEAP1?

KI696 is a potent (K_d ~1.3 nM) and highly specific non-covalent inhibitor of the NRF2-KEAP1 complex. Through its binding to the Kelch domain of KEAP1, KI696 disrupts KEAP1-mediated proteasomal degradation of NRF2, increasing NRF2 expression and transcription of its target genes (Davies et al. *J Med Chem* 59, 3991–4006 (2016)).

We have included the above summary in the revised manuscript (new text is underlined):

“To better understand the relationship between NRF2 activity and [¹⁸F]FSPG retention, we treated NRF2-low cell lines with the potent ($K_d \sim 1.3$ nM) and highly specific KEAP1 inhibitor KI696 (Fig. 2a). Through its binding to the Kelch domain of KEAP1, KI696 disrupts KEAP1-mediated proteasomal degradation of NRF2, increasing NRF2 expression and transcription of its target genes^{34, 35}”

4. The genetic makeup of the cells used in the experiments needs to be clarified. For example, does the STK11 status also change GSH accumulation and FSPG uptake?

Whilst other redox mechanisms, STK11 being one, may influence GSH accumulation and FSPG uptake, this is beyond the scope of this current work. Here, we demonstrate through the use of genetic, pharmacologic and advanced animal models, that NRF2/KEAP pathway activation perturbs redox homeostasis, which can be non-invasively identified by [¹⁸F]FSPG PET, with important therapeutic implications. It would be of interest to study competing redox mechanism in future work. We have indicated the plethora of competing redox control pathways as a limitation to the current work in the discussion section:

“Nutrient composition in the tumour microenvironment, metabolic alterations (e.g. anaplerosis), or other gene mutations (e.g. STK11) may therefore affect [¹⁸F]FSPG irrespective of NRF2 status.”

5. How sensitive is the visualization? Longitudinal follow-up of the signal and tumor quantification by CT needs to be graphed for the models shown in the main figures.

To address this important point, we performed new longitudinal [¹⁸F]FSPG imaging in NRF2-high H460 and NRF2-low H1299 subcutaneous tumours to address this important point. As the growth rates of these two tumours are quite different, we chose to image mice not as a function of time post-initiation but as a function of tumour size, starting off with very small volumes of 20-60 mm³. Even in these small tumours, there was a statistically significant difference in [¹⁸F]FSPG tumour retention between the two tumour types; a finding that was reproduced across all tumour sizes measured. At volumes >140 mm³ H460 tumours became necrotic, thereby reducing the average [¹⁸F]FSPG signal retained over the entire tumour volume. Clinically, simple SUV measurements of the single most intense voxel of a given volume will reduce issues related to average measurements. However, even in the >140 mm³ cohort, there was still a clear differential between NRF2-high and NRF2-low tumours.

Taken together, we believe these data further strengthen the evidence that PET imaging can differentiate NRF2 pathway activation in tumours. In-turn, this provides a non-invasive biomarker for treatment stratification.

The new data (Fig. 7d and Extended Figure 8) and associated text can be found below:

Figure 7. HM30-tesirine controls tumour growth and prolongs survival of mice bearing H460 tumours. **a**, Structure of the anti-xCT tesirine conjugate, HM30-tesirine. **b**, Western blot using HM30-tesirine as the primary antibody in H1299 and H460 cell lysates. Actin was used as a loading control. **c**, HM30-tesirine MTT dose-response in H460 and H1299 cells. **d**, Axial [¹⁸F]FSPG PET/CT images from mice bearing H460 or H1299 tumours before initiation of treatment. The dashed circle indicates the tumour. **e-f**, Antitumour activity (**e**) and survival benefit (**f**) of control (saline treated), cisplatin treated, and HM30-tesirine treated mice. The arrows under the x-axis of (**e**) represent treatment cycles for cisplatin (green) and HM30-tesirine (red). **g**, IHC for Ki67 and cleaved caspase 3 from FFPE tumours taken at endpoint. Scale bar, 50 μm. **h-i**, Corresponding quantification of tissue staining for Ki67 (**h**) and cleaved caspase 3 (**i**). Data are presented as the mean values ± SD (n = 3). **, p < 0.01; ***, p < 0.001.

Extended Figure 8. $[^{18}\text{F}]\text{FSPG}$ can distinguish NRF2-high from NRF2-low-expressing tumours across a range of tumour volumes. *a*, Representative axial $[^{18}\text{F}]\text{FSPG}$ PET/CT images of 40-60 min summed activity in mice bearing subcutaneous H460 (NRF2-high) and H1299 (NRF2-low) tumours of varying size. Dashed white lines indicate the tumour. *b*, Quantification of $[^{18}\text{F}]\text{FSPG}$ tumour retention. *, $p < 0.05$; **, $p < 0.01$; ***, $p < 0.001$.

Materials and methods:

“Longitudinal imaging of tumour xenografts

3×10^6 H460 cancer cells in 100 μL PBS or 5×10^6 H1299 cancer cells in 50 μL PBS, 50 μL Matrigel (Corning) were injected subcutaneously into female Balb/C nu/nu mice aged 6-9 weeks (Charles River Laboratories). Tumour growth was monitored using an electronic calliper, and the volume was calculated using the following equation: $\text{volume} = ((\pi/6) \times h \times w \times l)$, where h , w and l represent height, width, and length, respectively. Mice were longitudinally imaged by $[^{18}\text{F}]\text{FSPG}$ -PET across a range of tumour volumes, with thresholds set at 20-60 mm^3 , 60-100 mm^3 , 100-140 mm^3 and >140 mm^3 .

Results:

“We have shown above that $[^{18}\text{F}]\text{FSPG}$ measures NRF2 pathway activation with high sensitivity and specificity. Next, we performed longitudinal PET imaging in mice

bearing subcutaneous H460 or H1299 tumours with the aim of stratifying responders from non-responders. There was a marked and statistically significant difference in [¹⁸F]FSPG retention between these two tumour types (**Fig. 7d**). This pattern was replicated across all tumour sizes and stages of development, establishing that even in very small tumours (20 – 60 mm³), [¹⁸F]FSPG can discriminate NRF2 activation status (**Extended Fig. 8**). [¹⁸F]FDG, however, was unable to differentiate between NRF2-high and NRF2-low tumours (**Extended Fig. 9**).”

6. The signal from the GI seems different in each mouse and figure. Can the authors normalize the background signal or exposure? The signal in normal organs is quite high. How do the authors propose visualization for metastatic tumors? This should be stated as a limitation.

The imaging signal is thresholded differently for each figure to best represent the tumour FSPG retention for that particular model. Consequently, the healthy tissue signal appears to differ between the three animal models used. FSPG is one of the cleaner tracers developed: it is excreted through the kidneys, and the pancreas is the only healthy organ with high retention levels. This is what is seen in the GI and is clearly indicated in the figures and text, described below:

“PET imaging revealed a typical pattern of [¹⁸F]FSPG distribution, characterised by low physiological uptake in all healthy organs except the pancreas, and elimination via the urinary tract (**Fig. 3b** and **Extended Data Fig. 3**).”

Apparent variation in GI signal between mice is due to different imaging slices being selected to highlight the tumour, with these slices containing the pancreas to different degrees. Overall, pancreatic uptake isn't a problem when assessing metastatic disease; in the mouse, the pancreas is large and dispersed throughout the abdomen, whereas in humans, it is a discrete organ that traverses the kidneys. As is shown below, metastatic disease can be easily observed in NSCLC patients imaged with FSPG (Baek et al. (2012). J Nucl Med 54, 1-7; arrowhead points to the pancreas).

7. How does FDG glucose imaging look in these pairs of tumors? Is this a general metabolic uptake or specific to FSPG?

We performed FDG-PET imaging in size-matched H1299 (NRF2-low) and H460 tumours (NRF2-high) to answer this question. As expected, the two models did not differ in tumour FDG uptake, indicating that FDG is unable to identify NRF2 pathway activation.

We have created a new figure and described these findings in the revised manuscript, which is reproduced below:

Extended Figure 9. $[^{18}\text{F}]$ FDG cannot distinguish NRF2-high from NRF2-low-expressing tumours. Quantification of $[^{18}\text{F}]$ FDG tumour retention in mice bearing H460 (NRF2-high) and H1299 (NRF2-low) subcutaneous tumours. ns, not significant.

We have updated the manuscript's text as follows (new text is underlined):

“There was a marked and statistically significant difference in [¹⁸F]FSPG retention between these two tumour types (**Fig. 7d**). This pattern was replicated across all tumour sizes and stages of development, establishing that even in very small tumours (20 – 60 mm³), [¹⁸F]FSPG can discriminate NRF2 activation status (**Extended Fig. 8**). [¹⁸F]FDG, however, was unable to differentiate between NRF2-high and NRF2-low tumours (**Extended Fig. 9**).”

8. Imaging of the KP tumor is impressive, and the authors nicely complement this homogeneous model with the more heterogeneous PDX. What do the authors think about NRF2 activation in these mice versus actual tumors? How do the levels correlate?

Thank you for the compliment. Our TRACERx transcriptomic data shows that NRF2-regulated gene mRNA abundance (Fig. 5a) is high in approximately 1/3rd of patients. The best comparison we have between human tissue and our studies in mice are the two PDX xenografts that were derived from this patient cohort. mRNA abundance of NRF2-regulated genes in these tumours (Extended Fig. 5) is similar to those from patients (Extended Fig. 4). We have initiated a prospective clinical trial to assess the association between NRF2 activation and FSPG uptake, which will provide the ultimate proof in relation to your question.

9. In figure 4d, xCT expression seems not on the membrane. How is the ADC working? Also, how will the toxicity in normal tissues be addressed?

xCT protein is reported to localise to the membrane and to intracellular compartments like the lysosome, where there is continuous recycling (Mukhopadhyay et al. Proc Natl Acad Sci U S A. 2021 Feb 9; 118(6): e2021475118). However, our ADC targets only plasma membrane-localised xCT, resulting in remarkable efficacy (Fig. 7).

Our whole-body imaging data should predict normal tissue toxicity in the mouse. As xCT is only expressed at high levels in the pancreas, this is predicted to be the organ of dose-limiting toxicity. However, from our novel ADC is well tolerated at the dose used (Extended Fig. 10 and 11).

We have addressed the potential issue of toxicity in the Discussion section of the manuscript:

“System x_c^- , however, is not exclusively expressed on tumour cells, with high expression found in the brain, pancreas, and components of the immune system⁵³.⁵⁴. Although HM30-tesirine was well tolerated in mice, patients receiving xCT-targeted therapies should be monitored for on-target, off-tumour toxicity.”

10. In figure 5a the expression of the NRF2 gene signature appears low in most samples. Was that the conclusion from this data?

Approximately 1/3rd of patients had a high NRF2 gene signature, as shown in Fig.5a and Extended Fig.4. This is in line with previously published data (Cancer Genome Atlas Research. Comprehensive molecular profiling of lung adenocarcinoma. Nature 511, 543-550 (2014)). We have normalised these data and presented them in Fig. 5a as a z-score, which is the number of standard deviations away from the mean. As such, this, and possibly the colour pallet used, may give the false impression that the values in white and red are low, which is not the case.

11. Can this system detect small PDX tumors?

The tracer is agnostic to tumour type or model used, reflecting instead xCT activity. To address the question related to tumour size, we performed new longitudinal [¹⁸F]FSPG imaging in NRF2-high H460 and NRF2-low H1299 subcutaneous tumours. Even in small tumours, there was a statistically significant difference in [¹⁸F]FSPG tumour retention between the two tumour types; a finding that was reproduced across all tumour sizes measured. At volumes >140 mm³ H460 tumours became necrotic, thereby reducing the average [¹⁸F]FSPG signal retained over the entire tumour volume. Clinically, simple SUV measurements of the single most intense voxel of a given volume will reduce issues related to average measurements. However, even in the >140 mm³ cohort, there was still a clear differential between NRF2-high and NRF2-low tumours.

Taken together, we believe these data further strengthen the evidence that PET imaging can differentiate NRF2 pathway activation in tumours, regardless of the model used. In-turn, this provides a non-invasive biomarker for treatment stratification.

12. Does the xCT-ADC work on NRF2-low tumors?

In the revised manuscript, we performed new HM30-tesirine efficacy studies in the H1299 tumour model (NRF2-low). As would have been predicted both from the underlying biology and the FSPG imaging studies, HM30-tesirine had no efficacy in this model. Taken together, FSPG can, therefore, differentiate NRF2-high from

NRF2-low tumours with high sensitivity and specificity, which enables the stratification of subjects into responders and non-responders.

We have produced a new extended Figure that contains this new data and have adjusted the text accordingly (new text is underlined):

Extended Figure 11. HM30-tesirine has no efficacy in NRF2-low H1299 tumours. Anti-tumour activity (a), survival benefit (b) and animal weights (c) of control (saline treated), cisplatin treated, and HM30-tesirine treated mice. The arrows under the x-axis of a and c represent treatment cycles for cisplatin (green) and HM30-tesirine (red).

Results (p28):

“Conversely, HM30-tesirine did not affect tumour growth or survival in NRF2- and xCT-low H1299 tumours compared to vehicle control animals, whereas cisplatin induced a growth delay (**Extended Data Fig. 11**).”

Discussion (p32):

“Importantly, HM30-tesirine significantly extended the lives of H460 tumour-bearing mice compared to both cisplatin- and vehicle-treated animals. HM30-tesirine was ineffective in NRF2 WT tumours, highlighting the importance of [¹⁸F]FSPG in stratifying responders from non-responders.”

We hope that through these additional new in vivo experiments we have made a convincing case that FSPG imaging is a beneficial strategy to select subjects that are likely to benefit from our new xCT-targeted therapeutic.

REVIEWERS' COMMENTS

Reviewer #1 (Remarks to the Author):

and insightful comments. I think the authors have addressed several earlier concerns, but also some remain.

1) As shown in Figure 7 and implied by the text, the authors claim that FSPG PET predicts sensitivity to an Xct-targeted ADC. I am worried that using two cell line models is not sufficiently rigorous to make this claim. Certainly the data goes in the right direction, but it would be hard to convince a clinician to deploy this in a clinical setting based on the data of just two cell line models. Additionally, the factors that affect sensitivity of the biomarker in this setting are not elaborated. This is exacerbated by data showing that in some cases only a modest increase in FSPG is observed in some of the cell lines with activated NRF2 (some two fold or less). I think this should be more thoroughly explored.

2) Do the authors feel the ROC analysis is sufficiently powered?

3)The authors suggest that innate immune cells cleared caspase 3 positive tumors cells in treated mice in (now) Figure 7. I am very curious about this explanation because it appears immunodeficient mice were used for this experiment.

4) Regarding the PET study shown in Figure 7, the authors show that the FSPG low tumor and FSPG high tumor (baseline) respond differently. Ultimately the sensitive tumor progresses, it would be attractive to know whether the progression was driven by XCT/NRF2-mediated resistance and whether FSPG could demonstrate that. This would improve the importance of this potential biomarker.

Reviewer #2 (Remarks to the Author):

The authors have addressed all the major comments.

Reviewer #3 (Remarks to the Author):

Authors addressed majority of the comments while some remain unaddressed.

1-Authors can address comment #1 using RNA-sequencing data from different tumors (TCGA, TRACERx, etc.) to determine whether the connection between KEAP1 mutations and xCT expression holds true across other cancers.

2-Figure 7e/f: Clarify whether the tumors shown are lung or subcutaneous. If they are lung tumors, explain how the volumes were quantified, as the current legends do not provide sufficient information.

3-The limitations of background signal in PET-CT across different tissues, including the brain (as suggested by some of the images), should be mentioned. Additionally, the lack of clear membrane staining for xCT with the current method should also be noted.

REVIEWERS' COMMENTS

Reviewer #1 (Remarks to the Author):

and insightful comments. I think the authors have addressed several earlier concerns, but also some remain.

1) As shown in Figure 7 and implied by the text, the authors claim that FSPG PET predicts sensitivity to an Xct-targeted ADC. I am worried that using two cell line models is not sufficiently rigorous to make this claim. Certainly the data goes in the right direction, but it would be hard to convince a clinician to deploy this in a clinical setting based on the data of just two cell line models. Additionally, the factors that affect sensitivity of the biomarker in this setting are not elaborated. This is exacerbated by data showing that in some cases only a modest increase in FSPG is observed in some of the cell lines with activated NRF2 (some two fold or less). I think this should be more thoroughly explored.

Thank you for this point. This manuscript focuses on the ability for [¹⁸F]FSPG to provide a noninvasive marker of NRF2 status in models of NSCLC. We have shown the robustness of this measurement using genetic and pharmacologic manipulations in vitro and in GEMM, PDX, and orthotopic models in vivo, where the signal differential was 2-3-fold, which is more than sufficient for clinical imaging studies. Moreover, the ROC area under the curve approached 1, showing that the imaging readouts were both specific and sensitive.

We agree that the efficacy study, while convincing, was demonstrated in just two tumour xenografts, and so we have ensured that these data are not overinterpreted and highlighted the model used as a limitation in the revised text. This is reproduced below:

“Here, we provide proof-of-concept data that imaging system x_c^- may provide a means to select tumours most likely to respond to HM30-tesirine treatment. Care, however, should be taken in the interpretation of the resulting efficacy data, as the treatment was performed in a small number of xenograft models. Future work in a wider range of preclinical models is still required to demonstrate the connection between NRF2 activation and treatment efficacy prior to clinical translation.”

2) Do the authors feel the ROC analysis is sufficiently powered?

The ROC data is taken from 28, 168, and 20 tumours for the orthotopic, GEMM, and PDX studies, respectively. In our view, these data are well-powered for a preclinical study, which is reflected by the high statistical confidence of these analyses.

3) The authors suggest that innate immune cells cleared caspase 3 positive tumor cells in treated mice in (now) Figure 7. I am very curious about this explanation because it appears immunodeficient mice were used for this experiment.

Here, we used Balb/C nu/nu mice that only lack functional T-cells. The innate immune system (e.g. macrophages that will remove dead cells), and many features of the adaptive immune system are still intact.

4) Regarding the PET study shown in Figure 7, the authors show that the FSPG low tumor and FSPG high tumor (baseline) respond differently. Ultimately the sensitive tumor progresses, it would be attractive to know whether the progression was driven by XCT/NRF2-mediated resistance and whether FSPG could demonstrate that. This would improve the importance of this potential biomarker.

This is a great question, but one that is outside the remit of the current manuscript, which primarily focuses on imaging NRF2 by PET. We will follow up on this point in future papers.

Reviewer #2 (Remarks to the Author):

The authors have addressed all the major comments.

Thank you.

Reviewer #3 (Remarks to the Author):

Authors addressed majority of the comments while some remain unaddressed. 1-Authors can address comment #1 using RNA-sequencing data from different tumors (TCGA, TRACERx, etc.) to determine whether the connection between KEAP1 mutations and xCT expression holds true across other cancers.

The relationship between NRF2 and xCT is not new and is well-established, as shown by the direct binding of NRF2 to the antioxidant response element in SLC7A11's promotor and has been demonstrated in a range of cancer types.

The focus of this study, however, is NSCLC. The addition of different indications is certainly of great interest for us, but it will dilute the key message of the paper. We have added an additional line of discussion in the manuscript to highlight the role of NRF2 in other indications, which will be explored in future research:

“Furthermore, this antioxidant signature may be applicable to other cancer types with elevated NRF2, such as pancreatic ductal carcinoma⁴³.”

2-Figure 7e/f: Clarify whether the tumors shown are lung or subcutaneous. If they are lung tumors, explain how the volumes were quantified, as the current legends do not provide sufficient information.

We apologise for this omission. The tumours used were subcutaneous. We have further clarified how tumour volumes were quantified in the materials and methods and in the associated legend:

Legend

“e-f, Antitumour activity (e) and survival benefit (f) of control (saline treated), cisplatin treated, and HM30-tesirine treated mice bearing subcutaneous H460 tumours. n = 5-

6 mice per cohort. The arrows under the x-axis of (e) represent treatment cycles for cisplatin (green) and HM30-tesirine (red), with tumour volumes measured using electronic callipers.”

Materials and methods

“ 3×10^6 H460 cancer cells in 100 μ L PBS or 5×10^6 H1299 cancer cells in 50 μ L PBS, 50 μ L Matrigel (Corning) were injected subcutaneously into female Balb/C nu/nu mice aged 6-9 weeks (Charles River Laboratories). Tumour growth was monitored by electronic callipers as described above. When tumours reached ~ 70 mm³, mice were randomized into HM30-tesirine, cisplatin, and vehicle treatment groups. HM30-tesirine-treated animals received three doses of 1.5 mg/kg on days 0, 7 and 14 via an i.p. injection (200 μ L). The cisplatin-treated cohort received two doses of cisplatin (5 mg/kg; 200 μ L i.p) on days 0 and 4. Vehicle control animals were treated on days 0 and 7. Mice were weighed and tumour volumes measured until humane endpoints were reached (mean tumour diameter of 1.5 cm).”

3-The limitations of background signal in PET-CT across different tissues, including the brain (as suggested by some of the images), should be mentioned. Additionally, the lack of clear membrane staining for xCT with the current method should also be noted.

We have edited the manuscript to point out both xCT membrane and cytosolic staining.

“As expected, the Nrf2^{D29H/+} mutation also increased xCT compared to Nrf2^{WT} tumours, both in the membrane and the cytosol (**Fig. 4g,h**).”

Regarding FSPG background signal, FSPG does not pass the blood brain barrier. The signal which appears to be in the brain is from a 3D image (Fig. 5b) that picks up the salivary glands in the mouse from the same region of the skull. Salivary glands are not FSPG-avid in humans, so we do not see it as a limitation. We have, however, highlighted the issue in the revised discussion:

“Additionally, system x_c⁻, is not exclusively expressed on tumour cells, with high expression found in the brain, pancreas, and components of the immune system, which may provide confounding background signal in the PET images^{53, 54}”